METHODS AND RESOURCES

# Leveraging interindividual variability in threat conditioning of inbred mice to model trait anxiety

Irina Kovlyagina[1], Anna Wierczeiko[2], Hristo Todorov[2], Eric Jacobi[3], Margarita Tevosian[1,4], Jakob von Engelhardt[3], Susanne Gerber[2]*, Beat Lutz[1,4]*

1 Institute of Physiological Chemistry, University Medical Center of the Johannes Gutenberg University, Mainz, Germany, 2 Institute of Human Genetics, University Medical Center of the Johannes Gutenberg University, Mainz, Germany, 3 Institute of Pathophysiology, and Focus Program Translational Neuroscience (FTN), University Medical Center of the Johannes Gutenberg University, Mainz, Germany, 4 Leibniz Institute for Resilience Research (LIR), Mainz, Germany

* sugerber@uni-mainz.de (SG); beat.lutz@uni-mainz.de (BL)

**Data Availability Statement:** Individual values underlying the experimental data are provided in the S1 Data excel file. The RNAseq data were

## Abstract

Trait anxiety is a major risk factor for stress-induced and anxiety disorders in humans. However, animal models accounting for the interindividual variability in stress vulnerability are largely lacking. Moreover, the pervasive bias of using mostly male animals in preclinical studies poorly reflects the increased prevalence of psychiatric disorders in women. Using the threat imminence continuum theory, we designed and validated an auditory aversive conditioning-based pipeline in both female and male mice. We operationalised trait anxiety by harnessing the naturally occurring variability of defensive freezing responses combined with a model-based clustering strategy. While sustained freezing during prolonged retrieval sessions was identified as an anxiety-endophenotype behavioral marker in both sexes, females were consistently associated with an increased freezing response. RNA-sequencing of CeA, BLA, ACC, and BNST revealed massive differences in phasic and sustained responders' transcriptomes, correlating with transcriptomic signatures of psychiatric disorders, particularly post-traumatic stress disorder (PTSD). Moreover, we detected significant alterations in the excitation/inhibition balance of principal neurons in the lateral amygdala. These findings provide compelling evidence that trait anxiety in inbred mice can be leveraged to develop translationally relevant preclinical models to investigate mechanisms of stress susceptibility in a sex-specific manner.

## Introduction

*We like to think we have constrained an organism to the task we have set.*

*In practice, our paradigms are constrained by the way*

*the organisms respond*

McNaughton and Corr, 2004 [1].

deposited in NCBI's Gene Expression Omnibus (https://www.ncbi.nlm.nih.gov/pmc/articles/PMC99122/) and are accessible through GEO Series accession number GSE22269. All behavioural and electrophysiological data as well as the R code to reproduce the results reported in this study were deposited to Zenodo and are available at https://doi.org/10.5281/zenodo.10926698.

**Funding:** The author(s) received no specific funding for this work.

**Competing interests:** The authors have declared that no competing interests exist.

**Abbreviations:** AAC, auditory aversive conditioning; AAD, alcohol abuse disorder; ACC, anterior cingulate cortex; ASD, autism spectrum disorder; BD, bipolar disorder; BLA, basolateral amygdala; BNST, bed nucleus of stria terminalis; CR, conditioned response; CS, conditioned stimulus; DEG, differentially expressed gene; ECM, extracellular matrix; EPM, elevated plus maze; GAD, generalised anxiety disorder; GMM, Gaussian mixture model; HAB, high anxiety behaviour; IBD, inflammatory bowel disease; ISI, inter-stimulus interval; LA, lateral amygdala; LAB, low anxiety behaviour; LDT, light-dark test; LFC, log fold change; MDD, major depressive disorder; MR, memory retrieval; NOE, novel object exploration; OFT, open field test; PCA, principal component analysis; PNN, perineuronal network; PTSD, post-traumatic stress disorder; SCZ, schizophrenia; SNP, single-nucleotide polymorphism; STAI, state-trait anxiety inventory; TIC, threat imminence continuum; US, unconditioned stimulus.

Trait anxiety is a basic personality trait that refers to a consistent tendency of an individual to display an approach or avoidance behaviour in uncertain situations [1]. Personality and trait research in humans has identified trait anxiety as a major risk factor for developing mood and anxiety disorders [2]. People with high scores on the trait anxiety scale tend to experience negative emotions, such as nervousness, restlessness, irritability, and anxiety, more frequently and intensely. The World Health Organization estimated that 301 million people worldwide were affected by anxiety behaviour disorders in 2019. Therefore, understanding the neurobiology of trait anxiety has enormous public health significance, e.g., by leading to the discovery of new therapeutic targets (for definitions, see S1 Box). However, this necessitates the development of reliable preclinical models that facilitate the operationalization of trait anxiety [3–6].

To this end, several rat and mouse models based on the selective bidirectional breeding of animals according to their performance in unconditioned anxiety tests were established [7–10]. Selective breeding models have allowed the investigation of the genetic factors that contribute to the development and expression of trait anxiety. For example, using high and low anxiety behaviour (HAB and LAB) mouse lines, Kessler and colleagues linked a single-nucleotide polymorphism (SNP) in exon 1 of the vasopressin gene to the LAB phenotype of reduced anxiety-like behaviour [11]. Interestingly, different SNPs in the promoter region of the vasopressin gene contribute to an HAB phenotype in a rat model [12]. However, the limitations of selective breeding models for investigating neurobiological mechanisms are evident, as behavioural phenotypes cannot be directly replicated in transgenic lines. While selective breeding can establish strains with divergent anxiety-related behaviours, it does not allow for precise noninvasive manipulations of specific genes or molecular targets associated with anxiety. This limitation hampers the ability to dissect the causal relationships between specific genetic factors and anxiety-related behaviours. Apart from creating new lines, tests based on the approach-avoidance conflict are routinely used for assessing anxiety behaviour of rodents. Interestingly, the behavioural variability in these tests, which is fundamental to studying trait anxiety, is rarely acknowledged in animal research, possibly due to the assumption that using inbred lines is expected to reduce variability to a negligible minimum.

Nevertheless, observations conducted using long-term monitoring systems definitively confirmed that inbred animals display substantial individuality [13–15]. Therefore, animal models used in fear and anxiety research, particularly those aiming to understand human psychopathologies, must account for the behavioural variability in the experimental design and interpretation of results. Conversely, research only comparing the mean response of a stressed group to a control group of naïve animals has limited value for understanding human pathologies such as generalised anxiety disorder (GAD) and post-traumatic stress disorder (PTSD), which reflect extreme phenotypes that diverge from the population mean [16,17]. Indeed, an oversimplified approach to modelling behavioural responses may contribute to the lack of translational success of animal models and, as a result, the slow progress in the development of more effective anti-anxiety therapies [18,19].

Another challenge for translational research is the development of high-throughput behavioural testing pipelines that provide reliable and reproducible results. While longitudinal observation systems can provide a detailed characterisation of an individual animal, they are often impractical for the large-scale studies necessary for understanding of biological processes underlying behavioural variability. On the other hand, classical anxiety tests, such as open field test (OFT), light/dark preference test (LDT), and elevated plus maze test (EPM), require a novelty component, and when repeated using the same subjects, very low within-subject coherency is detected [20]. That leaves most of the experimental measurements cross-sectional, making it difficult to elucidate behavioural markers for classifying animals according to their traits.

In contrast, threat conditioning, a commonly used variation of classical conditioning [21] is a highly robust paradigm, where an initially neutral stimulus can be transformed into a threat signalling cue (conditioned stimulus, CS). The ability to associate cues with an upcoming danger is arguably the most important survival mechanism shared by all animals [22–25]. Upon CS recall, animals usually react with species-specific defensive behaviours drawn from an innate repertoire. Mice, as prey animals, typically initially become completely immobile (freeze) in the presence of a threatening stimulus. Importantly, since the conditioned response (CR) is remarkably stable across multiple retrievals, threat conditioning paradigms allow repeated testing, providing a powerful model for aversive learning and defensive behaviour investigation.

Similarly to anxiety tests, the variability of conditioned defensive freezing is usually not acknowledged as a factor in data analysis, despite multiple reports that animals exhibit significant heterogeneity in conditioned and unconditioned responses to threats [26–29]. Nevertheless, in recent years several mechanisms underlying interindividual differences in the expression of highly conserved conditioned freezing responses of rodents have been deciphered [30,31]. Authors of these studies classified animals into low- and high-fear groups by imposing cut-off criteria on the freezing response during memory retrieval. Yet, a detailed behavioural characterisation of low- and high-fear animals as well as translational interpretation of different freezing phenotypes were not the focus of these studies. Importantly, arbitrary cut-off values for low- and high-fear groups estimated from smaller samples might fail to generalise to independent cohorts.

In another, principally different approach, distinct anxiety phenotypes are elicited by subjecting animals to aversive conditioning paradigms with a predictable or unpredictable CS—unconditioned stimulus (US) timing [32,33]. These studies reported that rodents subjected to paradigms with predictable CS-US timing responded with phasic freezing to the onset of the CS during memory retrieval (MR) and returned to the baseline behaviour after 2 to 3 min before CS termination. In contrast, animals that went through the paradigms with unpredictable CS-US timing froze throughout the entire 6 min of CS presentation, shifting from phasic (first half of the CS) to sustained freezing (second half of the CS), representing a fear-potentiated anxiety behaviour. Therefore, authors of these publications argued for a primary role of learning in an animal's choice of a defensive response rather than individual differences in perception defined by (epi)genetics.

In the current study, we designed and validated an easy-to-adopt experimental pipeline combining a classical anxiety test battery and an auditory aversive conditioning (AAC) paradigm with prolonged retrieval sessions. We employed an unsupervised clustering algorithm to assign adult male and female C57BL/6J mice to 2 behavioural groups based on parameters of the freezing curves during memory retrieval. Subsequently, comparison of behavioural outcome measures from the anxiety test battery revealed that the obtained groups were indeed representative of 2 distinct trait anxiety endophenotypes (phasic and sustained responders) in both male and female mice subjected to an identical AAC protocol with unpredictable CS-US timing. Moreover, we uncovered vast differences in the transcriptomes of key brain regions in the defence circuit between phasic and sustained responders. Lastly, we followed up our findings by *ex vivo* electrophysiological recordings from neurons of the lateral amygdalar nucleus. In summary, we provide compelling evidence that freezing responses during threat memory retrieval reflect trait-like endophenotypes in inbred mice and thereby provide a measurable indicator that allows the operationalisation and in-depth investigation of trait anxiety.

## Results

### Variability of CRs unlocked by prolonging CS presentation during memory retrieval

To describe relationships between defensive behaviours and perceived proximity of a threat, an elaborate theoretical framework, the threat imminence continuum theory (TIC), was developed [34]. Depending on the spatial or temporal proximity of the threat or, more accurately, the perception of the threat along the imminence continuum, defensive behaviours can range from a defensive approach (risk assessment) to defensive avoidance (flight/freezing). In this study, we aimed to operationalise mouse anxiety trait using an experimental pipeline centred around the AAC paradigm by leveraging the main postulate of TIC that the imminence of a threat is a continuum defined by subjective perception.

Commonly used MR protocols usually consist of brief (30 s) CS exposures. Little variability in the CR expression is then observed due to the high predictive value of the cue—the imminence of the threat. Consequently, we sought to determine whether prolonged CS presentation during MR would cause individual animals' defence responses to unfold across the threat imminence continuum. In this case, we would be able to observe a diversity of defensive behavioural strategies defined by the threat perception of individual animals in response to an identical threatening stimulus. To this end, we utilised generic AAC protocols where aversive memories were retrieved by a 6-min CS presentation (see Methods). Retrieval sessions were analysed in 30-s time bins to capture the dynamics of the behavioural response.

First, we tested the influence of CS-US timing during conditioning on the freezing response during MR using published protocols (S1A and S1B Fig). The goal of this experiment was to determine whether CS-US timing manipulation during AAC can condition the entire cohort of subjected animals uniformly to either a phasic or sustained response during retrieval as suggested by Daldrup and colleagues [32]. Although we detected significant differences in the average freezing response between mice conditioned using predictable and unpredictable protocols (S1C Fig), the variability of CR in both animal cohorts was similar. Consequently, the heterogeneity of individual freezing responses allowed us to classify animals into phasic and sustained responders independently of CS-US timing during training (S1D Fig). Since we observed a slightly increased average freezing response in mice conditioned using the AAC protocol with unpredictable CS-US timing, we chose it for further investigation of the interindividual variability of CR and its association to anxiety trait.

Both male and female wild-type C57BL/6J mice conditioned to the tone (Fig 1A), responded with uniformly high freezing to the immediate presentation of the CS during MR (Fig 1B and 1C). However, the initially low variance in freezing responses steadily increased, reaching a maximum value 2.5 min after CS onset (time bin 18) and remained at high levels thereafter (Fig 1D). Accordingly, the freezing curve during prolonged CS exposure can be segmented into 2 components (Fig 1E). During the phasic component (bins 13–17), all animals reacted to CS with strong freezing due to the high predictive value of the cue. In the sustained phase (bins 18–24), highly individual responses emerged with some animals returning to baseline values while others retained high levels of freezing (Fig 1B and 1C). We observed a significantly increased freezing response in females compared to males in 2 independent cohorts of animals (Figs 1F and S1E). The largest differences between sexes were detected during the sustained freezing phase where the individuality of defence response strategies was most pronounced. Remarkably, we either did not observe any differences in locomotion (Fig 1G) between males and females prior to conditioning or females were moving more than males (S1F Fig).

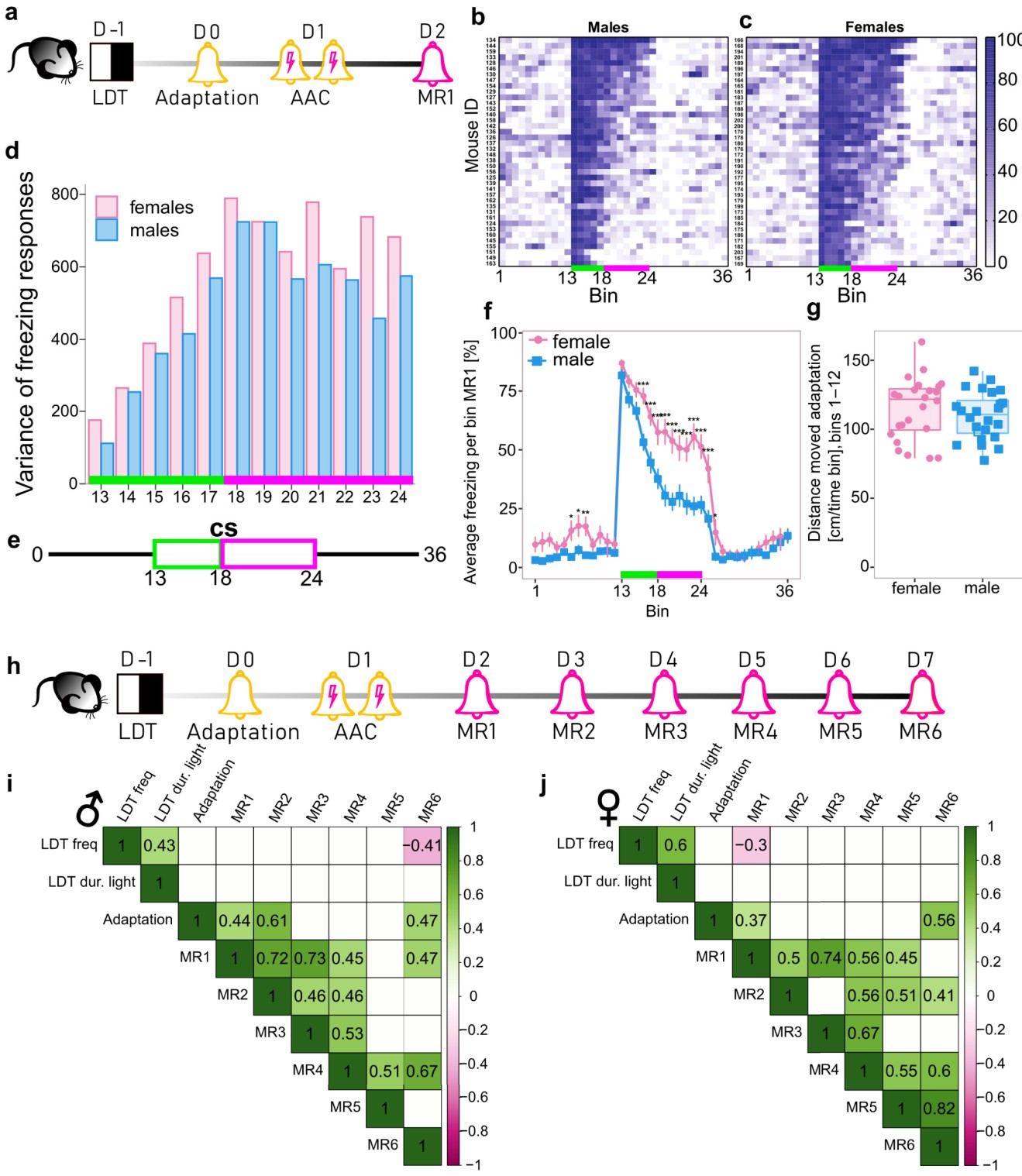

**Fig 1. Variability of CRs unlocked by prolonging CS presentation during MR. (a)** Experimental timeline. **(b, c)** Representative heatmaps showing individual freezing response (%) in males (b) and females (c) during MR1 in 30-s time bins. CS bins are indicated with a green/magenta line (bins 13–24). **(d)** Variability of freezing responses during CS presentation. Bar plots show the variance of freezing in each time bin for male and female mice. **(e)** Schematic representation

of the MR protocol. CS bins are indicated with green/magenta line corresponding to the phasic and sustained components of freezing, respectively. **(f)** Females (n = 24) have a higher average freezing response than males in MR1 (n = 24). **(g)** Comparison of distance moved between females and males during the adaptation day prior to conditioning (n = 24 per sex). **(h)** Experimental timeline for multiple memory retrievals. **(i, j)** Correlation matrices of behavioural metrics for males (i) and females (j). Only Pearson correlation coefficients with p < 0.05 are shown (i, j). For LDT, Adaptation, and MR1, data from 2 independent batches were pooled together (n = 48/per sex), for MR2-MR6 (n = 24/per sex). Adaptation = freezing during time bins 13–14 during the adaptation day, MR1-MR6 = average freezing from time bins 18–24 from each MR session. All mice were conditioned using the protocol with unpredictable CS-US timing. *p < 0.05, ** p < 0.01, ***p < 0.001, linear mixed effects model followed by pairwise comparison of model means between sexes in (f), unpaired t test in (g). Individual values underlying the experimental data are provided in the S1 Data excel file. CS, conditioned stimulus; LDT, light/dark test; MR, memory retrieval; US, unconditioned stimulus.

To determine whether the freezing response during MR is a phenotype-like characteristic, we conducted an LDT prior to conditioning and performed 5 additional MRs on consecutive days (Fig 1H). We observed strong positive correlations of sustained freezing during MRs (Fig 1I and 1J) indicating consistency/stability of this readout. Interestingly, phasic freezing (bins 13–17) underwent significant extinction from MR1 to MR6 as indicated by the negative coefficients in a linear regression model, confirming that phasic freezing reflects a memory component (S1G and S1I Fig). At the same time, the extinction rate of the sustained component (bins 18–24) did not change significantly over time in both sexes (b = −1.26, p = 0.08 in females and b = −1.81, p = 0.062 in males, S1H and S1J Fig), indicating that while sustained freezing is also a conditioned response, its variability is driven by other factors—most likely differences in innate trait. Moreover, in 4 independent cohorts of animals (2 per sex), we assessed the stability of the freezing response after 1 month (Fig 2A). Based on the strong correlations observed in these experiments (Figs 1I, 1J, 2I, and S3B), we hypothesised that sustained freezing responses during prolonged retrieval sessions reflect the animal's anxiety state and can be used as a behavioural marker of an anxiety endophenotype.

## Identifying phasic and sustained responders

Prompted by our observation that individual animals have unique freezing curves during CS presentation that likely reflect distinct anxiety states, we next aimed at developing a computational strategy for identifying sustained and phasic responders from the total population of animals. To this end, we employed a series of log-linear regression analyses to obtain a mathematical model of each animal's freezing response during CS presentation (Fig 2B, Methods). Subsequently, we used the intercept and slope (corresponding to the freezing starting value and the rate of freezing decay, respectively) of the fitted curve together with average sustained freezing (bins 18–24) during MR1 to cluster animals into 2 groups—phasic and sustained responders (Figs 2C, S2F, and S2G). Freezing during bins 18–24 was included in our model based on the empirical observation that variability of freezing responses was highest during the sustained phase in both sexes (Fig 1D). Therefore, this parameter strongly reflects the individuality of freezing behaviour. Mice clustered to the phasic group were characterised by a faster decay of the fitted curves and average sustained freezing comparable to baseline values or unconditioned animals' responses to CS. In contrast, sustained responders had a lower decay rate and average sustained freezing significantly higher than baseline (Figs 2D, 2E,–S2H, and S2I).

Furthermore, in cohorts subjected to the extended phenotyping pipeline, we repeated the clustering procedure with MR2 data. Using this strategy, we identified consistent phasic and sustained responders that were assigned to the same phenotype in both clustering iterations. However, a proportion of animals (27.5% to 34%) shifted from a sustained to a phasic phenotype indicating safety learning. Additionally, a fraction of animals (15.8% to 17.5%) changed their assignment from the phasic to the sustained group corresponding to threat generalisation

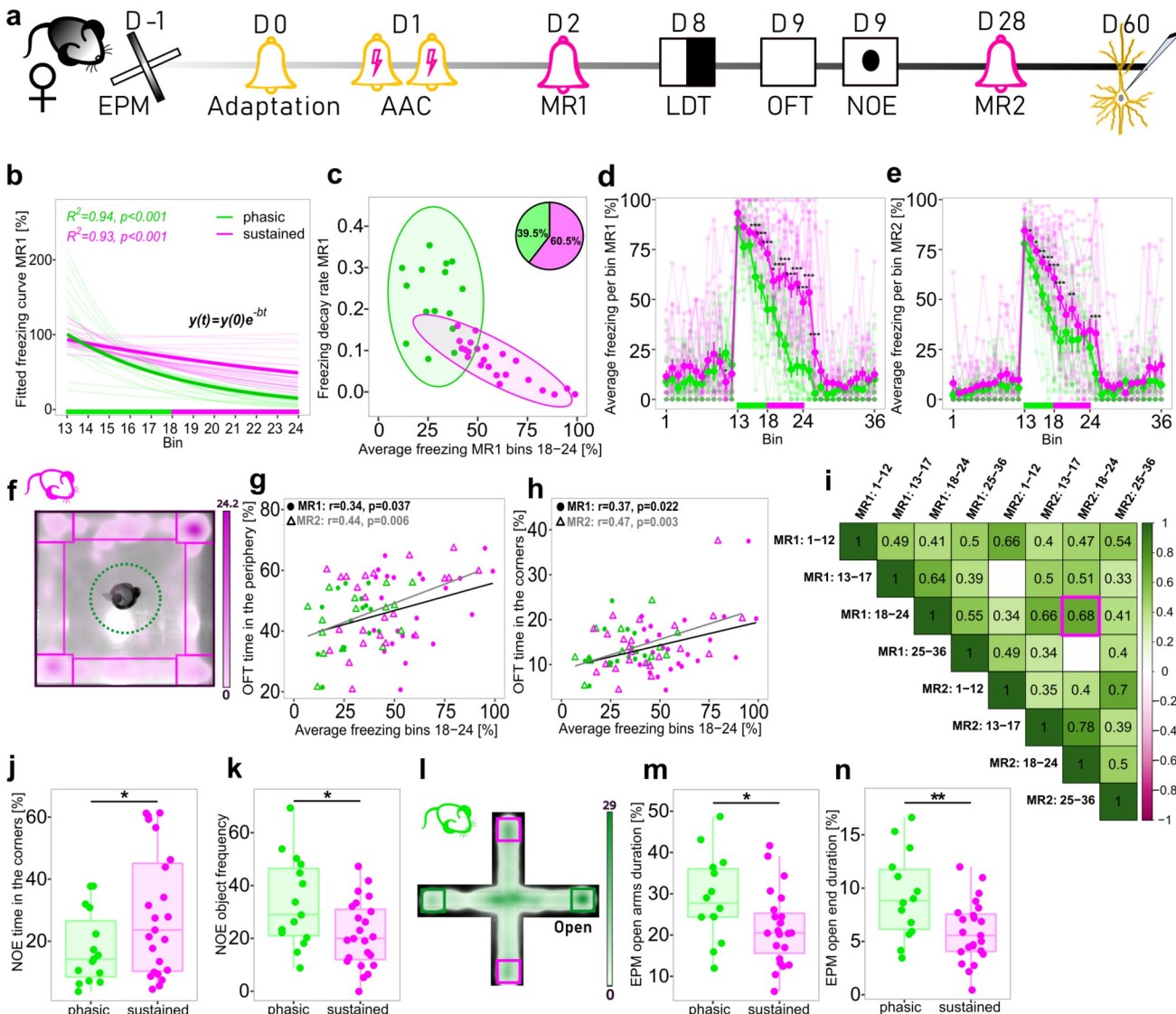

**Fig 2. Identification and behavioural characterisation of phasic and sustained responders.** (a) Experimental timeline. (b) In pale, fitted curves modelling individual freezing response during CS presentation (bins 13–24) using log-linear regression models, in bold fitted curves based on the group average of each endophenotype in females during MR1. (c) Clustering of a single batch of female mice into phasic (green) and sustained (magenta) responders based on average sustained freezing (bins 18–24), decay rate, and intercept of the freezing curve during MR1; 95% confidence ellipses around each cluster are shown. (d, e) In pale, individual freezing curves, in bold, average freezing curves of each endophenotype during MR1 (d) or MR2 (e). (f) Arena settings of OFT and NOE overlaid on the sustained responder's representative heatmap. (g, h) Pearson correlation plots of sustained freezing during MR1 and MR2 with thigmotaxis behaviour during OFT. (i) Correlation matrix of different phases of MR1 and MR2. Only Pearson correlation coefficients with p < 0.05 are shown. (j) Sustained responders spent more time in the arena's corners during the NOE test. (k) Sustained responders approached the novel object less frequently during the NOE test. (l) Arena settings of EPM overlayed on the phasic responder's representative heatmap. (m) Sustained responders spent less time in the open arms during the EPM test. (n) Sustained responders spent less time in the ends of open arms during the EPM test. n = 38 in all panels. All mice were conditioned using the protocol with unpredictable CS-US timing. *p < 0.05, **p < 0.01, ***p < 0.001, linear mixed effects model followed by pairwise comparison of model means between sustained and phasic responders in d and e, unpaired t test in j, k, m, n. Individual values underlying the experimental data are provided in the S1 Data excel file. CS, conditioned stimulus; EPM, elevated plus maze test; MR, memory retrieval; NOE, novel object exploration test; OFT, open field test; US, unconditioned stimulus.

behaviour. Collectively, we referred to these 2 subgroups as "shifters" (S2C, S3E, and S5B Figs). These results demonstrate that individual freezing responses can be harnessed to subdivide the overall population of conditioned animals into consistent phasic and sustained

responders as well as animals that change their coping strategies after repeated CS exposure. Remarkably, the proportion of animals assigned to each group was stable and comparable across independent animal cohorts (S3D and S5C Figs).

## Is sustained freezing a behavioural marker of anxiety trait?

We did not identify any systematic phenotype or sex-dependent differences in freezing response during aversive training (S2A–S2D Fig), indicating that the heterogeneity in sustained freezing did not emerge as a consequence of differences in memory acquisition. Therefore, we hypothesised that sustained freezing is reflective of the animal's anxiety trait. To investigate this hypothesis, we included classical anxiety tests such as EPM, LDT, OFT, and novel object exploration (NOE) tests in our behavioural pipeline (Fig 2A). Presenting simple approach–avoidance tasks, these tests were chosen to assess the animal's intrinsic conflict sensitivity–anxiety trait [1]. We intentionally selected a redundant testing strategy since traits, defined as consistent behavioural patterns within an individual across various contexts, necessitate evaluation through diverse tasks for effective classification. We detected significant correlations between sustained freezing during MR1 and MR2 with thigmotaxis behaviour in OFT (Fig 2F–2H). After placing a novel object in the middle of the open field arena, we observed that sustained responders spent significantly more time in the corners of the arena (Fig 2J) and approached the novel object less frequently than phasic responders (Fig 2K). Remarkably, both groups of animals already behaved differently prior to conditioning. Phasic responders showed an increased risk assessment behaviour in the EPM test compared to sustained responders as indicated by the total duration in the open arms (Fig 2L and 2M) and on the open ends of the arena (Fig 2N). Taken together, these results strongly suggest that the anxiety state during MR is a derivative of the animal's anxiety trait (see S1 Box).

## Selection of phasic and sustained responders for transcriptomic profiling

To substantiate the claim that our behavioural observations constitute real endophenotypes driven by molecular differences between phasic and sustained responders, we performed a transcriptomic analysis of key brain regions in the defensive circuit 24 h (MR1) and 28 days (MR2) after conditioning. For this analysis, we selected phasic and sustained responders with the most consistent phenotypes. In the batch with next day retrieval only (MR1), we chose 12 animals per sex with extreme phasic and sustained freezing profiles, indicated by low sustained freezing/high decay rate and high sustained freezing/low decay rate, respectively (S2E–S2J Fig). In the batch subjected to the extended phenotyping pipeline (S3A Fig), we first assigned animals to a phenotype using data from MR1 (S3C Fig) and then repeated the clustering procedure using measurements from MR2 (S3D Fig). Animals that were assigned to the same phenotype in both clustering iterations were considered high-confidence and retained for further selection (S3E–S3G Fig). Then, we leveraged performance in the OFT to identify candidates with a consistent approach or avoidance behaviour strategy and selected 6 phasic and 6 sustained female responders that spent the shortest/longest time in the periphery of the arena, respectively, for transcriptomic profiling (S3H–S3L Fig). Additionally, selected phasic responders had a strong tendency (p = 0.08) to spend more time exploring the light compartment in the LDT (S3M Fig).

Anterior cingulate cortex (ACC), basolateral (BLA), central amygdalar nuclei (CeA), and bed nucleus of stria terminalis (BNST) were sampled by a tissue punching technique from brains harvested 30 min after onset of CS during MR1 or MR2. These brain regions have been implicated as central regions in threat conditioning and anxiety circuits in numerous studies [35–39].

In female samples collected after MR1, we identified 391 differentially expressed genes (DEGs) in ACC, 143 DEGs in BLA, 3,030 DEGs in CeA, and 248 DEGs in BNST (p. adj < 0.05) (Figs 3A, 3B, S4A, and S4B and S1 Table). Furthermore, there were 2 DEGs in ACC, 253 DEGs in BLA, 493 DEGs in CeA, and 124 DEGs in BNST (p.adj < 0.05) in brains collected after MR2 (Figs 3C and S4D–S4F and S1 Table).

In male samples collected after MR1, we identified 2 DEGs (p.adj < 0.05) in ACC (S4C Fig) and 38 DEGs (p.adj < 0.05) in CeA (Fig 3D and S2 Table). Principal component analysis (PCA) revealed larger sex-dependent differences in CeA than in ACC (S4G Fig). Due to the low number of DEGs in male animals in these 2 brain regions after MR1, we did not include additional brain regions and sequencing after MR2 for male responders.

## Transmembrane transporter genes consistently up-regulated in BLA of female phasic responders but acutely down-regulated in CeA

Among the 292 DEGs, overlapping between MR1 and MR2 (Fig 3E), we detected a group of 13 genes consistently down-regulated in BLA of phasic compared to sustained responders but up-regulated in CeA during MR1 (Fig 3F and 3G). Notably, Ttr (encoding transthyretin) that had the largest log fold change (LFC) in all 3 conditions was accompanied by Otx2 (Orthodenticle Homeobox 2), which was previously shown to regulate Ttr expression [40]. Kcne2 (voltage-gated potassium channel accessory subunit 2), Kcnj13 (potassium inwardly rectifying channel), *Aqp1* (aquaporin 1), *Slc16a8* (lactate transporter), *Slc4a5* (sodium bicarbonate cotransporter), *Folr1* (folic acid receptor and 5-methyltetrahydrofolate transporter), *Steap1* (six-transmembrane epithelial antigen of prostate), *Tmem72* (transmembrane protein 72), and *Ecrg4* (augurin) had the same expression pattern. Two other DEGs in BLA overlapping between MR1 and MR2 samples: *Cdh3* (cadherin 3 or calcium-dependent adhesion protein) and *Kl* (klotho) are also transmembrane proteins. In contrast, from the 110 overlapping DEGs in CeA, only 3 were consistently up-regulated in phasic responders. The other 107 DEGs had the opposite LFC during MR2 compared to MR1 (Fig 3F). The same pattern of expression was observed for DEGs detected in BNST at both time points (Fig 3F). For example, *Pvalb* (parvalbumin) was increased almost 10-fold in phasic responders during MR1 but after MR2 it was decreased 32-fold. Interestingly, *Pvalb* was also up-regulated in CeA of phasic responders during MR1, and 62 more genes were differentially regulated in both CeA and BNST during MR1 most of which had the same direction of regulation (Fig 3E). In addition, a PCA revealed that CeA and BNST were separated from BLA and ACC along the first principal component (S4H and S4I Fig), indicating a higher similarity between subcortical and cortical structures, respectively, and thereby confirming the accuracy of our sampling technique.

From 38 genes differentially expressed in CeA (p.adj < 0.05) of male phasic and sustained responders at MR1, 8 overlapped with female CeA DEGs (Fig 3D). Two of these genes, *Nexn* (nexilin) and *Scn4b* (sodium channel, voltage-gated, type IV, beta), were also more than 18-fold up-regulated in the BLA of female phasic responders during MR1 as compared to sustained responders. Additionally, *Nexn* was more than 4-fold down-regulated in BNST after MR2.

Synaptic plasticity genes were down-regulated in the CeA of female phasic responders 24 h after conditioning. Among the unique CeA DEGs, we identified a group of important regulators of development and maturation of glutamatergic neurons and synapses: *Tbr1* (T-box brain transcription factor 1), *Bdnf* (brain derived neurotrophic factor), *Neurod2* (neuronal differentiation 2), and *Neurod6* (neuronal differentiation 6, NEX). Interestingly, glutamate transporter *Slc17a7* (VGLUT1) was also substantially down-regulated in CeA of phasic responders. *Neurod2* and *Neurod6* belong to the neural lineage of the basic-helix-loop-helix

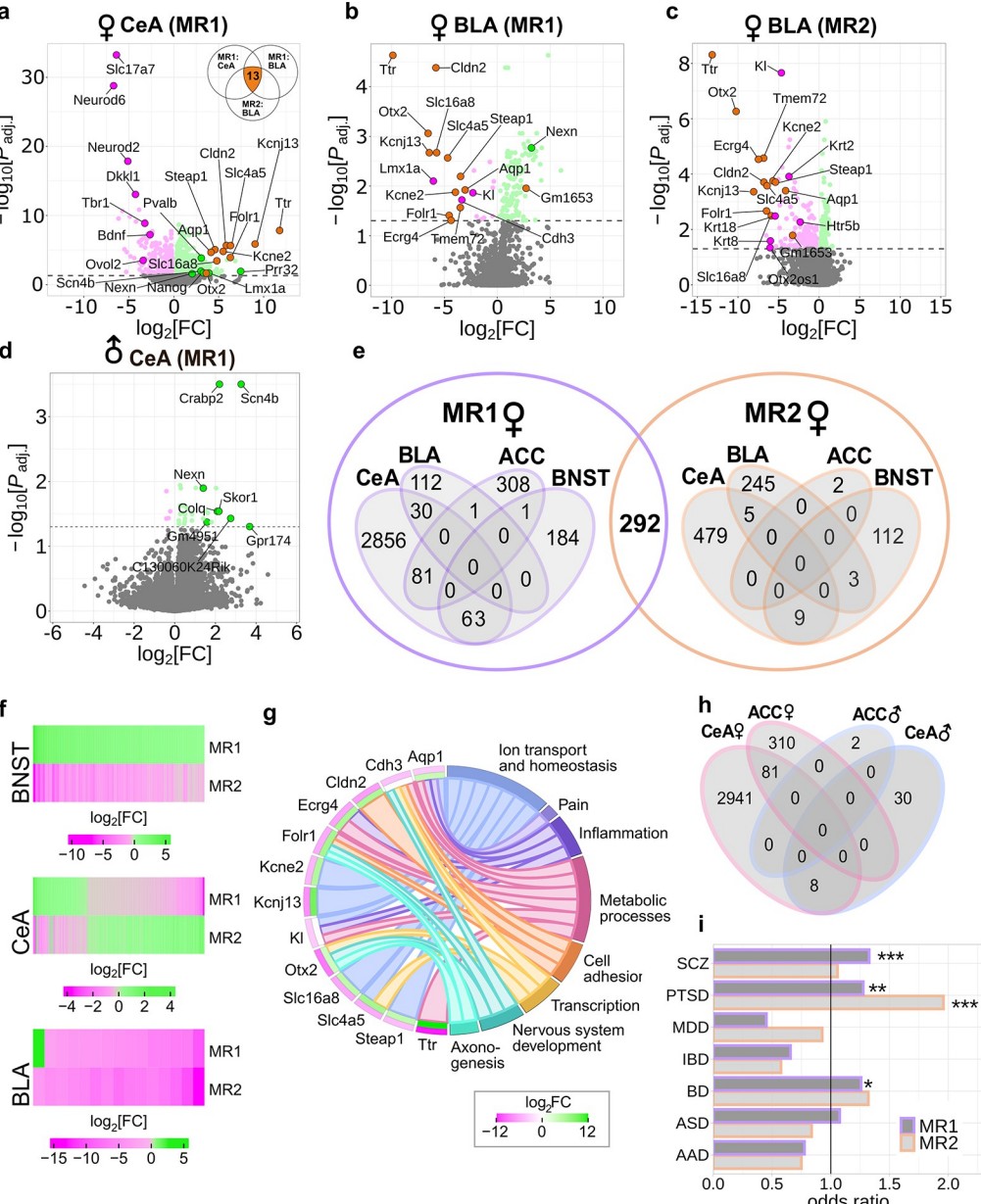

**Fig 3. Transcriptomic profiling of phasic and sustained responders. (a–c)** Volcano plots showing log2-fold change against adjusted p-value of all expressed genes in selected brain regions of phasic versus sustained female responders. DEGs (adjusted p-value <0.05) are coloured in green and magenta for up- (LFC > 0) and down-regulation (LFC < 0) in phasic compared to sustained responders, respectively. **(d)** Volcano plot showing log2-fold change against p-value of expressed genes in the CeA of phasic versus sustained male responders after MR1 (adjusted p < 0.05). Positive log2-fold changes indicate genes that were up-regulated in phasic compared to sustained responders. **(e)** Venn diagrams showing the intersection of DEGs (adjusted p-value <0.05) between all sequenced female brain regions. **(f)** Heatmaps illustrating BNST, CeA, and BLA overlapping DEGs between MR1 and MR2 in female mice. Positive log2-fold changes indicate genes that were up-regulated in phasic compared to sustained responders. **(g)** Chord plot depicting 15 genes that were differentially expressed in CeA (MR1), BLA (MR1), and BLA (MR2) of female mice and their annotation to selected biological processes (S10 Table). Positive log2-fold changes indicate up-regulation in phasic compared to sustained responders. The outer layer shows the average log2-fold change for each gene in BLA (MR1 and MR2) and the inner layer shows the log2-fold change in CeA (MR1). Positive and negative average LFCs are coloured in green and magenta, respectively. **(h)** Venn diagram showing the overlap between DEGs (adjusted p-values <0.05) in the sequenced male brain regions and DEGs (adjusted p-values <0.05) from matching female conditions. **(i)** Psychiatric disorders-related gene enrichment analysis. Odds ratios significantly greater than 1 indicate an enrichment of the corresponding disease-related gene set in the DEGs in our study. *p < 0.05, **p<0.01, ***p < 0.001, Fisher's exact test in i. All mice were

conditioned using the protocol with unpredictable CS-US timing. AAD, alcohol abuse disorder; ASD, autism spectrum disorder; BD, bipolar disorder; BLA, basolateral amygdala, CeA, central amygdala; CS, conditioned stimulus; DEGs, differentially expressed genes; IBD, inflammatory bowel disease; MDD, major depressive disorder; MR, memory retrieval; PTSD, post-traumatic stress disorder; SCZ, schizophrenia; US, unconditioned stimulus.

(bHLH) family of transcription factors and their co-expression in postnatal brain with *Slc17a7* has been previously shown [41]. *Tbr1* was also down-regulated together with its target *Fezf2* (FEZ family zinc finger 2) [42]. These findings further support the validity of our sequencing results. Notably, *Sst* (somatostatin) and *Cort* (cortistatin, SST2) were also down-regulated in the CeA of female phasic responders. The co-regulation of these 2 genes with *Bdnf* had been previously established and linked to major depressive disorder (MDD) [43,44].

## Transcriptomes of phasic and sustained female responders significantly overlap with transcriptomic signatures of human psychiatric disorders

In addition to alterations in *Bdnf*, *Ttr*, and *Neurod2* expression implicated in MDD and autism spectrum disorder (ASD), respectively [45,46], we detected some other genes implicated in psychiatric disorders. For instance, *En2* (engrailed 2), an ASD-associated gene [47,48], was up-regulated in ACC of phasic responders more than 5-fold. *Slc6a4* (serotonin transporter), which is associated with a number of psychiatric conditions [49,50], was 2-fold down-regulated in CeA of phasic responders. Therefore, we performed a comparative analysis between the transcriptomic signatures of phasic and sustained female responders and the transcriptomes of cerebral cortex tissue collected *post mortem* from humans with major psychiatric disorders such as PTSD [51], ASD, MDD, alcohol abuse disorder (AAD), bipolar disorder (BD), schizophrenia (SCZ), and inflammatory bowel disease (IBD), which was used in the original study as non-neural control [52] (Fig 3I and Material and methods). This analysis revealed a significant overlap of DEGs detected in our study with genes deregulated in PTSD (both at MR1 and MR2), SCZ (MR1 only), and BD (MR1 only) patients (Fig 3I and S12 Table). In particular, out of the 591 PTSD-related DEGs in the human studies, 144 overlapped with MR1 and 53 with MR2 DEGs between phasic and sustained responders. Furthermore, 501 of the 2,044 SCZ-associated genes were also differentially expressed in phasic and sustained responders at MR1. For BD, we detected an overlap of 125 out of the 518 DEGs in the human cohorts with our data (S12 Table).

## Functional enrichment analysis revealed acute inflammation in CeA of female phasic responders

To further characterize transcriptomic signatures of phasic and sustained responders, we performed functional enrichment analysis of differentially expressed genes using gene ontologies (GOs), KEGG, and REACTOME databases (Material and methods). DEGs were analysed either all together or up- and down-regulated separately. In all conditions, joint and separate analysis yielded similar results (S3–S5 Tables), except CeA at the early time point (MR1). In this region, up- and down-regulated sets of DEGs were enriched for inflammation and synapse organisation terms, respectively (Fig 4C). In line with these findings, cell type enrichment analysis of up- and down-regulated DEGs in CeA revealed enrichment for genes specific for non-neuronal cell types among the up-regulated and for neuron-specific genes in the down-regulated DEGs (Fig 4B). At the late time point (MR2), we did not detect inflammation related terms in CeA (Fig 4A), suggesting that inflammation-like processes were transient and triggered by aversive learning. Related to inflammation, extracellular matrix (ECM) organisation-associated DEGs were also up-regulated in CeA at MR1 (Fig 4C). Surprisingly, inflammation-

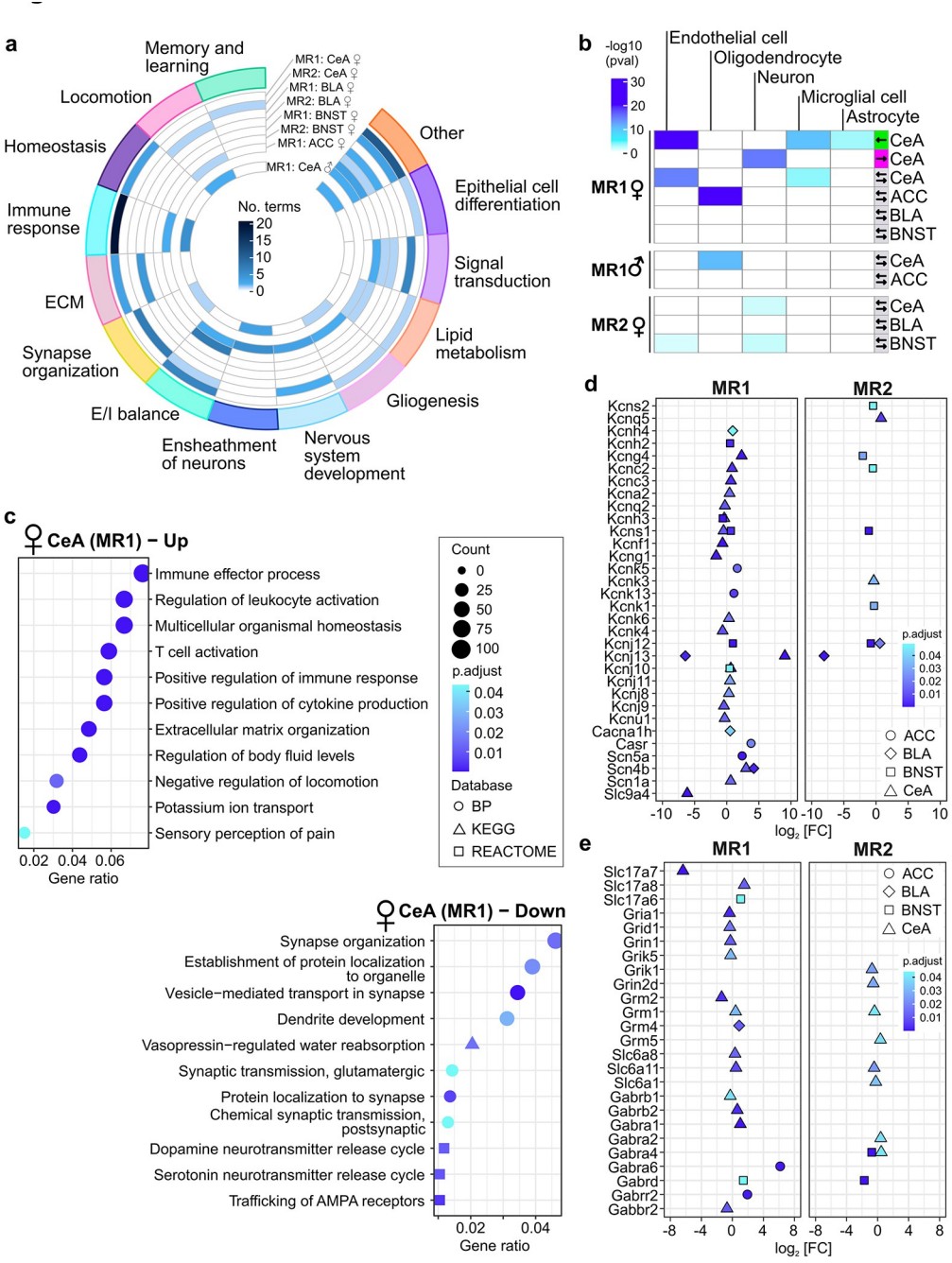

**Fig 4. Functional enrichment analysis. (a)** Circos plot summarising functional enrichment analysis (biological process GO terms assigned to higher level categories) using combined up- and down-regulated DEGs of females identified in each condition, and males in CeA at MR1. **(b)** Cell type enrichment analysis for the DEGs in females (adjusted p-value < 0.05) and genes with raw p-value < 0.05 in males. Coloured cells correspond to statistically significant cell-type enrichment results (p < 0.05), and arrows at the right indicate the direction of regulation of input genes. **(c)** Dot plot showing functional enrichment analysis of up- or down-regulated DEGs detected in CeA (MR1) of female mice. **(d)** Genes encoding for a broad variety of potassium, calcium, and sodium channels differentially expressed in phasic compared to sustained responders in female mice. **(e)** Glutamatergic and GABAergic synaptic transmission-related genes differentially expressed in phasic compared to sustained responders in female mice. Fisher's exact test in b. BLA, basolateral amygdala; BNST, bed nucleus of stria terminalis; CeA, central amygdala; DEGs, differentially expressed genes; MR, memory retrieval.

related DEGs were up-regulated, while synapse organisation DEGs were down-regulated in phasic compared to sustained responders. In contrast, at the 28-day time point (MR2), synapse organisation terms were enriched for up-regulated genes in phasic responders.

Consistent with the behavioural output, BLA, CeA, and ACC transcriptomes (MR1) were enriched for genes regulating locomotion. ACC (MR1) DEGs were enriched for ECM and inflammation-related genes as well (Fig 4A and 4C). BNST showed consistent enrichment for potassium channels encoding genes although the direction of regulation was opposite in MR2 compared to MR1.

To power functional enrichment analysis of male transcriptomes statistically, we included DEGs with an unadjusted p-value < 0.05. Similar to CeA in females, we detected multiple terms related to inflammation, synapse organisation, and locomotor behaviour (Fig 4A and S6–S8 Tables) suggesting that differential expression of many genes in male CeA might have not reached significance due to higher behavioural variability of males. Interestingly, again inflammation and ECM-related DEGs were up-regulated in phasic responders (S7 Table).

## Altered excitability of primary neurons in lateral amygdala of sustained responders

Our analysis of DEGs also revealed that multiple genes, coding for a broad variety of ion channels, as well as essential components of the glutamatergic and GABAergic signalling machinery, are differentially expressed between female phasic and sustained responders in all the regions at both time points (Fig 4D and 4E). Hence, we investigated basic electrophysiological properties, action potential-related firing, as well as AMPA and GABA receptor-mediated synaptic currents of principal neurons in the lateral amygdala (LA). We selected this region for 2 reasons. First, the LA is a central hub of aversive learning-associated plasticity. Second, from the investigated brain regions, we observed consistent alterations in gene regulation only in the BLA (Fig 3F). Additionally, many of the BLA DEGs overlapping between MR1 and MR2 are reportedly involved in neuronal signalling key processes such as action potential firing and excitatory and inhibitory synaptic transmission (Fig 3G). Based on these findings, we postulated that intrinsic electrophysiological properties, action potential firing, and/or excitatory/inhibitory synaptic transmission might be altered in phasic compared to sustained responders. Importantly, we hypothesised that the differences in electrophysiological properties between the 2 phenotypes would be subtle and fall within the distribution observed in wild-type mice in previous studies that have not differentiated between subphenotypes as we did.

Using the strategy described above, we selected 8 phasic and 8 sustained female responders to perform whole-cell patch-clamp experiments in acute brain slices of the LA (Figs 2A–2N and S5A–S5E). Selected animals showed robust approach or avoidance phenotypes, not only during both retrievals and OFT, but also in the EPM test prior to conditioning (S5F–S5K Fig). Patch-clamp recordings were performed 2 months after AAC, thus our data more likely reflect the basic properties of the investigated cells rather than short-term effects of the behavioural testing.

We found no differences in the analysed intrinsic electrical properties (resting membrane potential and input resistance) of the primary neurons in the LA between phasic and sustained responders (Fig 5A–5D). However, the threshold of action potential generation was significantly increased in phasic compared to sustained responders (Fig 5E), while all other investigated action potential- and firing-related parameters showed no differences between the 2 endophenotypes (Figs 5F, 5G, and S6A–S6E). The analysis of the synaptic miniature currents in principal cells of the LA showed that, while there were no detectable differences in the AMPA receptor-mediated mEPSC properties (Figs 5I–5K and S6F), amplitudes and risetimes

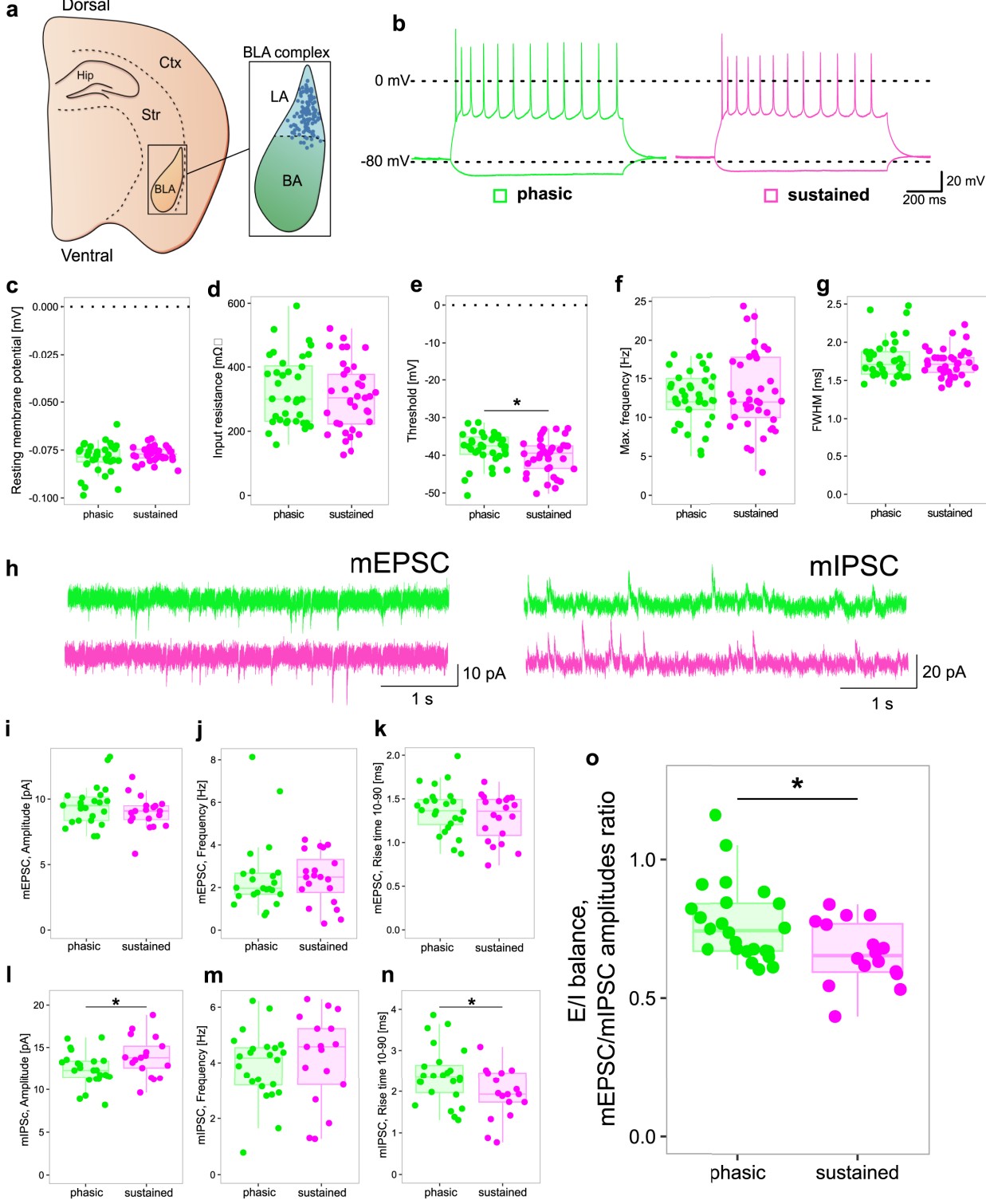

**Fig 5. Electrophysiological properties of lateral amygdala principal neurons. (a)** Schematic illustration of the location of the BLA complex in an acute horizontal slice. The blue dots represent the location of the single neurons within the BLA, which were recorded in the patch-clamp experiments. **(b)** Sample traces of the firing pattern of primary LA neurons from sustained (magenta) or phasic (green) responders. **(c, d)** Analysis of intrinsic electrical properties of primary LA neurons. Resting membrane potential (c) and input resistance (d) are not altered between primary LA neurons of sustained and phasic responders (n = 37 in phasic and n = 37/38 in sustained groups) **(e–g)**. Analysis of action potential related

properties. The threshold of action potential generation is increased in primary LA neurons of phasic responders compared to the same cell type in sustained responders (e). Frequency (f) and action potential kinetics (g) are not altered between primary LA neurons of sustained and phasic responders. (n = 36/37 in phasic and n = 38 in sustained groups). (**h**) Sample traces of AMPAR-mediated mEPSC and GABAR-mediated mIPSC recordings of primary LA neurons of sustained and phasic responders. (**i–k**) No differences in mEPSC properties were detected between phasic and sustained responders (n = 24 in the phasic and n = 19/20 in the sustained groups). (**l–n**) mIPSC events of primary LA neurons of phasic responders show a reduced amplitude (l) and an increased rise time (n) compared to neurons from sustained responders. The frequency (m) of the mIPSC events is unaltered between the 2 groups (n = 24 in phasic and n = 17 in sustained group). (**o**) Analysis of the excitation inhibition ratio (E/I ratio) of primary neurons of the LA of both phenotypes. Sustained responders showed a stronger inhibitory drive onto primary neurons of the LA compared to the same cells in phasic responders (n = 24 in the phasic and n = 16 in the sustained groups). *$p < 0.05$, unpaired t test. Individual values underlying the experimental data are provided in the S1 Data excel file. BLA, basolateral amygdala; LA, lateral amygdala; mEPSC, miniature excitatory postsynaptic currents; mIPSC, miniature inhibitory postsynaptic currents.

of GABA receptor-mediated mIPSCs were decreased and increased, respectively (Figs 5L–5N and S6G).

In order to better assess the effect of these changes on the neuronal network of the LA, we analysed the E/I ratio (Fig 5O) of each recorded cell. Indeed, we observed that the increase in GABA receptor-mediated currents leads to an increased inhibitory drive on principal neurons in the LA of sustained compared to phasic responders. Therefore, our results confirmed differences between phasic and sustained responders in several key aspects of neuronal signalling, such as action potential threshold and inhibitory synaptic signalling. This might reflect the differential expression of many genes involved in neuronal signalling pathways that we detected before and, hence, contribute to the establishment of two different behavioural endophenotypes in mice.

## Discussion

In the current study, we developed a phenotyping pipeline consisting of a behavioural test battery and a model-based clustering approach that enables the detection and in-depth profiling of mice with different anxiety endophenotypes. By combining a redundant behavioural testing strategy with a high-throughput explorative correlational analysis [53], we identified a behavioural marker of anxiety trait—the magnitude of sustained freezing during prolonged MR. This parameter appeared to be a generalizable characteristic across sexes and multiple animal batches acquired from different breeding facilities (S9 Table).

A central finding of our study was the correlation of sustained freezing with baseline freezing during MR as well as with approach–avoidance behaviour during anxiety tests conducted before and after conditioning. We thereby confirmed that sustained freezing represents an anxiety state that is not independent but, in fact, derivative from the anxiety trait [54,55]. In contrast to previous work relying on time-consuming multigenerational breeding, we confirmed that high- and low-anxiety behaviour endophenotypes can be reliably isolated within 1 generation of inbred mice. Therefore, this pipeline can be easily applied to any transgenic line, circumventing the main limitation of selective breeding approaches.

Moreover, our proposed pipeline can be viewed as a translational analogue of state-trait anxiety inventory (STAI) used to investigate the interaction of trait anxiety with state anxiety in humans [56]. Such an animal model could provide crucial insights into biological mechanisms controlling susceptibility to stress and anxiety disorders, as the anxiety trait is a significant contributor to interindividual differences in stress vulnerability in humans [2]. Indeed, we detected a significant overlap of transcriptomic signatures of phasic and sustained responders with genes deregulated in psychiatric conditions such as PTSD, SCZ, and BD (Fig 3I and S12 Table).

Contrary to previous reports [33,57], we observed heterogenous freezing responses during MR independent of the conditioning protocol. Although the unpredictable CS-US timing

Figure 6

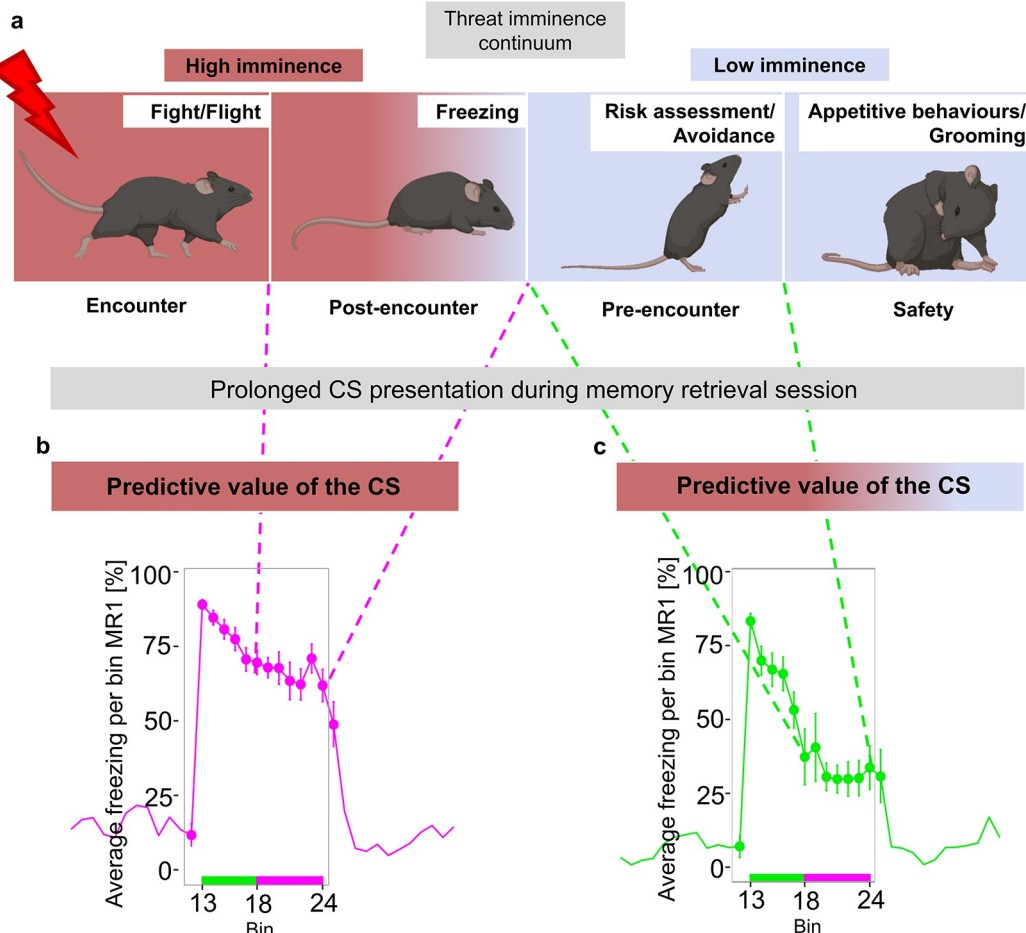

**Fig 6. Individual changes in perceived predictive value of the CS during prolonged presentation elicit distinct defensive strategies along the threat imminence continuum. (a)** Different defensive modes across the different stages of threat imminence. While predictive value of CS remained high throughout presentation for sustained responders (b), it changed for phasic responders shifting animals' behaviour from post-encounter freezing to pre-encounter risk assessment (c). CS, conditioned stimulus.

during conditioning resulted in a slightly increased average freezing compared to a predictable protocol, both configurations produced fundamentally similar ranges of individual freezing responses. Therefore, manipulating only CS–US timing does not strongly affect the animal's defensive strategy, and thus, likely, CS reinforcement rate during conditioning plays a more important role in estimating the imminence of the threat during retrieval. In contrast, manipulating the CS presentation length during MR enabled the exploration of a range of defensive behaviours in mice along the threat imminence continuum. This strategy allowed us to capture not only the initial freezing response–avoidance behaviour—corresponding to the proximal end of the threat imminence continuum, but additionally harness the response when the threat perception is placed in the middle and far end of the continuum. Our results indicate that, throughout presentation, the CS predictive value changes differently for each animal depending on the individual's anxiety trait (Fig 6).

While we operationalised anxiety through observable freezing behaviour, it is important to acknowledge that freezing is not a direct measure of anxiety trait, but merely a manifestation of an animal's behavioural coping strategy in a particular testing context. Therefore, a trait can also be viewed as a vector of coping strategies, meaning that multiple points of assessment are necessary to make conclusions about an animal's trait. Consequently, animals exhibiting consistent coping strategies, (phasic and sustained responders) coexist alongside those who adjust their coping strategy in case of repeated exposure to the same testing context (shifters). Irrespective of endophenotype assignment, freezing behaviour was on average a very consistent measure, as demonstrated by strong positive correlations observed between MR1 and MR2 in several independent animal cohorts (Figs 1I, 1J, 2I, and S3B). However, further investigation into the characteristics and implications of animals with a shifting coping strategy is necessary to understand the full spectrum of anxiety endophenotypes.

Another important issue we addressed in our study are sex-dependent differences in patterns of defensive behaviour and their underlying biological substrate. While women are affected by anxiety disorders 2 times more often than men [58,59], neuroscience research has frequently neglected females in the study design [60,61]. However, recent investigations involving both sexes have shown significant differences in conditioned freezing responses and underlying mechanisms [30,62–64]. Interestingly, in studies using a contextual aversive conditioning paradigm, sex-dependent differences in CR were observed during contextual MR [64], while after AAC training, differences were detected only between animals with extreme phenotypes when memories were retrieved using a 30-s CS [30]. Consistent with these observations, we did not detect any differences between males and females during the first 30 s of the CS presentation (S1M Fig), as well as during training (S2B–S2D Fig) suggesting no sex-dependent differences in memory acquisition. The most prominent differences occurred during the sustained component of freezing (Figs 1F and S1E). Moreover, after classifying animals according to their anxiety trait, we saw that while phasic responders of both sexes behaved similarly, sustained female responders spent significantly more time in the periphery of the open field arena compared to the matching male group (S1N Fig). Additionally, we observed much more pronounced differences between behavioural endophenotypes in female mice (S2H–S2J Fig). This effect might be partially attributable to our clustering strategy that produced more distinct subpopulations in females because of a more even distribution of freezing responses along the whole spectrum, while male freezing responses showed an overall skewed distribution towards lower freezing (S1K and S1L Fig). Consequently, the effect size of freezing response differences was much greater in sustained versus phasic females compared to males (Cohen's d for average sustained freezing (bins 18–24) = 3.27 in females versus 1.47 in males in the batch of animals used for RNAseq experiments at the MR1 time point). This observation could be explained by different roles each sex plays in the survival of the species. In the wild, male mice have to protect the territory from intruders and, therefore, have to be biased towards approach in threatening situations [65]. On the other hand, female mice that nurture their offspring and forage in the wild have to select task-specific behavioural strategies.

In line with the larger differences in freezing behaviour of females, transcriptomic analysis of brain structures controlling the defensive strategy choice (ACC and CeA) [37,66] revealed much larger gene expression differences between phasic and sustained females than between phasic and sustained males. Therefore, we focused on the in-depth molecular profiling of females and included into the analysis 2 additional brain regions implicated in aversive learning and anxiety behaviour (BLA and BNST). Although in the present study we did not aim to investigate the exact molecular mechanisms involved in shaping an animal's anxiety trait, our transcriptomic findings are in line with multiple published studies. For instance, vasopressin, which was previously implicated as one of the key regulators of HAB and LAB phenotypes

[67], was also 16-fold up-regulated in BLA of phasic compared to sustained responders further confirming that identified freezing phenotypes are representative of anxiety trait. Moreover, the largest differences were detected in CeA between phasic and sustained female responders 24 h after conditioning. About 10% of all the genes were differentially expressed in this condition. The circuit mechanisms controlling conditioned freezing responses have been thoroughly investigated. Current evidence supports the theory that mutually inhibitory internal circuits of CeA are gating the transition between different defensive modes using a winner-takes-all strategy (for an excellent review on the role of CeA in threat perception see [68]). Therefore, one possible mechanism explaining differences in sustained freezing responses could be the interplay of mutually inhibiting neuronal subtypes SOM+ and PKC-δ+ [37,69]. Indeed, our results support this as potential mechanism, as *Sst* (somatostatin), *Cort* (cortistatin, SST2), and *Sstr 2*, *3*, and *4* (somatostatin receptor types 2, 3, and 4) were all found to be differentially expressed in the CeA of female phasic and sustained responders 24 h after conditioning. Of course, further experiments using transgenic mouse lines to manipulate specific cell types will be required to validate this hypothesis.

Another major finding of our transcriptomic profiling is a group of 15 genes encoding transmembrane proteins differentially expressed in BLA at both time points. These genes have been previously implicated in a diverse spectrum of biological processes including ion transport and homeostasis (Fig 3G). Moreover, functional annotation of CeA DEGs identified the ECM organisation pathway as one of the top significant terms (Fig 4C and S4 Table). Altogether, these findings may suggest that while a switch between SOM+ and PKC-δ+ neurons in CeA can acutely regulate the transition between defensive modes, differences in ECM composition and turn-over can contribute to shaping the anxiety trait in the long-term by providing a distinct environment for developing and mature brain cells, thereby influencing synaptic plasticity. Remarkably, differences in BLA ECM perineuronal networks (PNNs) of 2 mouse strains with robust differences in extinction have been previously described, suggesting that divergent behaviour phenotypes might have origins in early development [70]. Indeed, it has been shown that BLA PNNs form during development as a mechanism shielding aversive memories from extinction-induced plasticity [71]. Additionally, the observed differences in tests preceding AAC, such as LDT (Fig 1Js) and EPM (Fig 2L–2N) further strengthen the argument for the developmental origins of anxiety endophenotypes.

One more transcriptomic finding indirectly supporting the notion of developmental origins of trait anxiety was a group of 25 genes encoding bHLH transcription factors differentially regulated between phasic and sustained responders in all conditions. Seven of these genes belong to the neural lineage clade of the bHLH superfamily. Although these transcription factors are particularly important during development, it has been shown that *Neurod2* regulates calcium signalling and homeostasis of mature neurons [72] as well as spine turnover in the postnatal cortex [45]. *Neurod2* and its paralog *Neurod6* were 64-fold down-regulated in phasic responders. Remarkably, a haploinsufficient phenotype of *Neurod2* had been shown previously in multiple studies [45,73]. Lin and colleagues reported that heterozygous *Neurod2* knock-out mice have a decreased freezing response during both contextual and cued memory retrievals and an increased approach behaviour in anxiety tests [73]. In another study, overexpression of *Neurod2* in ventral hippocampus induced a stress susceptible profile [74]. Interestingly, *Dkkl1*, which was validated *in vivo* in the same study as increasing stress susceptibility, was also down-regulated in phasic responders in our data set (Fig 3A).

Importantly, changes in expression of *Neurod2/6* [41,75], *Dkkl1* [74], and *Bdnf* [76], have been linked to altered electrophysiological properties of affected neurons. Nevertheless, the large total number of differentially expressed genes we identified (>3,000) made it impossible to predict which specific properties would be altered in individual cells or networks associated

with different behavioural phenotypes. Additionally, behavioural traits are polygenic in nature and regulated by large networks of neurons from different brain regions. Based on this rationale, we performed recording of basic electrophysiological properties of principal neurons in LA.

Among all recorded parameters, we detected decreased firing threshold of principal neurons in LA of sustained responders 2 months after conditioning. Additionally, the ratio of the amplitudes of mEPSCs to mIPSCs was significantly larger in phasic than in sustained responders, indicating that inhibition of principal neurons in phasic responders is lower than in sustained responders. The reduction in E/I balance in "sustained" neurons (Fig 5O) may appear counterintuitive to the higher freezing observed in this group. However, the observed changes in the E/I balance likely have more complex effects on neuronal excitability and spiking, which may depend on the specific pattern of synaptic inputs and outputs, as well as the expression and isoform of different ion channels and postsynaptic receptors.

Overall, our results confirmed differences between phasic and sustained responders in several key aspects of neuronal signalling, such as action potential threshold and inhibitory synaptic signalling. These findings may reflect either intrinsic differences in major components of neuronal signalling of phasic and sustained responders or changes induced by the aversive stimulus. Indeed, previous research has shown that aversive stimuli can trigger neuroinflammation in the amygdala [77], which can alter the electrophysiological properties of neurons, contributing to hyperactivity of amygdala and, in the end, manifesting as an anxiety disorder [78]. Further research is needed to fully understand the molecular adaptations in the defence circuit that underlie different anxiety endophenotypes.

In essence, our behavioural pipeline for identifying phasic and sustained endophenotypes based on operationalisation of trait anxiety, that was developed and intensively characterised over 9 batches and a total of 243 animals, has a number of potential applications and implications for research in several areas. For example: (i) translational studies seeking to understand biological mechanisms of stress susceptibility and anxiety disorders to accelerate drug discovery can benefit from a robust phenotyping tool; (ii) basic research on memory formation and synaptic plasticity could use this classification to investigate how differences in trait anxiety may influence aversive learning; and (iii) our pipeline could be integrated in studies of neurodevelopmental processes to assess the impact of genetic mutations or other experimental prenatal manipulations on distribution of trait anxiety-dependent behaviour. Overall, our study contributes to the ongoing paradigm shift in behavioural research towards deciphering individual traits and bridging the gap to translational research.

## Materials and methods

### Animals

All experiments were performed according to the European Community's Council Directive of 22 September 2010 (2010/63EU) and approved by the Landesuntersuchungsamt of the State Rhineland-Palatinate, Germany; file numbers of ethical approvals: 23 177-07/G16-1-085, 23 177-07/G17-1-028, and 23 177-07/G21-17-028. C57BL/6J mice were purchased from the Charles River and Janvier breeding facilities (for animal assignment, see S1 Table) at the age of 8 to 9 weeks and allowed to habituate to the new facility for at least 3 weeks. Mice were group-housed in temperature- and humidity-controlled rooms with a 12-h light-dark cycle with water and food provided ad libitum. Seven days prior to behavioural experiments, animals were single-housed. All mice reached 12 weeks of age at the beginning of testing.

Behavioural tests were carried out during the light phase, trials were video recorded and analysed with Noldus Ethovision XT software, version 13 (Noldus, Wageningen, The

Netherlands). After each trial, the set-ups were cleaned with water (all tests) or 1% acetic acid after auditory aversive conditioning.

## Elevated plus maze (EPM)

EPM was performed using a cross-shaped set-up (Noldus, Wageningen, The Netherlands) having 2 open and 2 closed arms elevated 100 cm above the floor. The arms of the maze were 35-cm long and 6-cm wide. Experimental mice were placed into the maze facing the closed arms and were allowed to freely explore for 10 min. Animals were video recorded and tracked automatically. For zoning of the arena, see Fig 2L. A mouse was considered to be in the zone if the center point was in the zone.

## Light-dark test (LDT)

Experimental animals were placed into a custom box (39 × 39 cm), divided into the light zone (two thirds of the box, white walls) and dark zone (one third of the box, separated and covered by a 26-cm high lid, black walls). The light and dark compartments were connected with a small entry zone (5 × 5 cm). Animals were placed in the dark compartment and allowed to explore freely and move between the dark and light zones for 10 min. Time spent in the light zone (center point) was assessed using video recording and subsequent automated analysis. A mouse was considered to be in the zone if the center point was in the zone.

## Open field test (OFT)/Novel object exploration (NOE)

Mice were placed individually in an open white box (40 × 27 × 40 cm) and allowed to explore for 5 min (OFT). Then, animals were briefly removed from the box to the individual home cages, and the object was placed in the middle of the arena. Afterwards, mice were returned to the arenas and allowed to explore the object and arena for another 10 min. The exploration zone was defined as a circular zone 2 cm in diameter around the object. Animals were video recorded and tracked automatically. For zoning of the arena, see Fig 2F. A mouse was considered to be in the zone if the center point was in the zone, except for the object exploration zone where the nose-point was used.

## Auditory aversive conditioning (AAC)

All days of the AAC paradigm were performed using TSE Multi Conditioning chambers, series 256060 (TSE Systems GmbH, Bad Homburg, Germany). Animals were video recorded and tracked automatically offline using Ethovision 13 (Noldus, Wageningen, The Netherlands). Briefly, during the **Adaptation** session (day 0), mice were habituated to context A (cylinder wrapped in nontransparent paper, electric grid covered by the white plastic panel, fresh bedding was used for each mouse, chambers were cleaned between trials with water). Individual mice were placed in cylinders, after 6 min of exploration mice were presented with a Tone (6 min/10 kHz, 75 dB), then mice were left in the chambers for 6 more minutes. On the next day (day 1), mice went through AAC in context B (transparent rectangular, electric grid not covered, chambers cleaned between trials with 1% acetic acid). The 2 AAC protocols used in this study were previously described in Daldrup and colleagues, **Predictable protocol** (Extended data, Fig 1A): 2 min habituation, 4xCS (18 s/10 kHz, 75 dB), inter-stimulus interval (ISI) 15 s, 19 s, 20 s; each CS was co-terminated with 1 s 0.4 mA electric foot shock. After the last shock, mice were left in the chambers for 30 s. Training was repeated in 5 h. **Unpredictable protocol** (Extended data, Fig 1B): 2 min habituation, 4xCS (29 s, 9 s, 19 s, 15 s/10 kHz, 75 dB), ISI 30 s; each CS was co-terminated with 1 s 0.4 mA electric foot shock. After the last

shock, mice were left in the chambers for 30 s. Training was repeated in 5 h with altered CS order (14 s, 19 s, 9 s, 29 s). In the **No-shock protocol**, CSs were not reinforced by the foot shock. **Memory retrieval** (MR) was performed 24 h later (day 2) in context A (cylinder wrapped in nontransparent paper, electric grid covered by the white plastic panel, fresh bedding was used for each mouse, chambers cleaned between trials with water) using an identical protocol from Adaptation day. All subsequent MRs were done using the same protocol. For Adaptation day, 0.5 s and for MRs, 1 s freezing thresholds were applied for final data analysis. To further separate conditioning and retrieval contexts, these phases were performed by different people. Mice were not specifically habituated to experimenters prior to the experiments. There was no significant experimenter effect in either female ($F(1,122) = 1.764$, $p = 0.187$) or male animals ($F(1,85) = 0.511$, $p = 0.477$) on the sustained freezing response as evaluated by an ANOVA analysis across multiple animal cohorts.

## Isolation of brain regions of interest

Mice were sacrificed by decapitation 30 min after the last MR, brains were snap-frozen on dry ice and stored at −80˚C. Regions of interest (ACC, BNST, BLA, and CeA) were isolated by a brain punching technique from the whole-mount brain in the cryostat Leica CM3050 S (Leica Biosystems, Nussloch, Germany) at −20˚C using biopsy pens EMS Rapid-Core (Electron Microscopy Sciences, Hatfield, United States of America). To localise the subregions of interest in the brain, punching was carried out by referencing toluidine blue staining of brain slices [79]. ACC was sampled approximately from 1.34 to 0.5 mm (from bregma) using a 1 mm pen. BNST (0.5, −0.22) mm, BLA (−0.94: −2.06) mm, and CeA (−0.94: −1.82) mm were sampled bilaterally using a 0.5 mm pen. Biopsies were stored at −80˚C for further RNA extraction.

## RNA isolation

All steps were carried out on ice. Tissue biopsies were moved to precooled Precellys tubes containing ice-cold ceramic beads for homogenisation via Precellys 24 (Peqlab, Erlangen, Germany) (600 g, 20 s). Extraction was performed using RNeasy mini kit (Qiagen, Hilden, Germany) according to the protocol provided by the manufacturer.

## RNA sequencing

The quality of RNA was assessed via Bioanalyzer (Agilent Technologies Germany GmbH & Co. KG, Waldbronn, Germany) and quantity was assessed via Qubit 4 (Invitrogen, Life Technologies GmbH, Darmstadt, Germany). mRNA was isolated from 100 ng of total RNA (individual samples). Libraries were prepared using NEBNext Ultra II Directional RNA library preparation kit for Illumina (New England Biolabs GmbH, Frankfurt am Main, Germany). Libraries were sequenced to the depth of at least 60 million reads. Female and male samples after MR1 were sequenced (single-end reads, $1 \times 75$ nt) with Illumina NextSeq 500 (Illumina, San Diego, USA). Female samples after MR2 were sequenced (single-end reads, $1 \times 50$ nt) with Illumina NextSeq 2000 (Illumina, San Diego, USA). Quality checks and sequencing were performed by StarSEQ GmbH (Mainz, Germany).

## Electrophysiology

Brains were removed from deeply anesthetised (3% isoflurane) and cardially perfused 20- to 28-week-old mice. Cardiac perfusion was done with ice-cold PIPES solution (in mM: Pipes 20, Sucrose 200, KCl 2.5, NaH2PO4 1.25, CaCl2 0.5, MgSO4 10, Glucose 10), and 250 to 300 µm coronal slices of the lateral amygdala were cut in 4˚C PIPES solution (see above) perfused with

95% O2 and 5% CO2 (pH 7.4). Freshly cut brain slices were kept for half an hour in 32°C warm Ringer solution (in mM: NaCl 125, NaHCO3 25, KCl 2.5, NaH2PO4 1.25, CaCl2 2, MgCl2 1, Glucose 25) and later stored in room temperate Ringer solution for a minimum of 30 min.

Whole-cell recordings were performed at room temperature using pipettes pulled from borosilicate glass capillaries with a resistance of 3 to 5 MΩ. Pipettes were filled for firing pattern experiments with an intracellular solution containing (in mM): K-gluconate 130, Hepes 10, Phosphocreatin-Na 10, Na-Gluconate 10, ATP-Mg-Salt 4, NaCl 4, GTP 0.3. For mEPSC/mIPSC recordings, the intracellular solution contained (in mM): Cs-Methanesulfonate 120, MgCl2 5, Hepes 30, Phosphocreatine-Na 10, EGTA 0.6, ATP-Mg-Salt 4, GTP 0.4. Liquid junction potentials were not corrected. In mEPSC/mIPSC recordings, series resistance and input resistance were continuously monitored by measuring peak and steady-state currents in response to small hyperpolarizing pulses. mEPSCs were recorded holding the cell at −70 mV and mIPSCs at −10 mV. Slices were continuously perfused with Ringer (see above) saturated with 95% O2/5% CO2 (pH 7.4). For mEPSC/mIPSC recordings, TTX and AP-V were added to the bath solution.

## Behavioural data analysis

Behavioural data analysis was performed using the R language and environment for statistical computing version 4.1.2.

We employed regression analysis to derive a mathematical model of each animal's freezing response during CS presentation (bins 13–24). Based on the actual freezing curves, we postulated that freezing behaviour follows an exponential decay model:

$$y(t) = y(0)e^{-bt},$$

where y(t) corresponds to freezing during time bin t, y(0) indicates the freezing starting value (intercept), and b corresponds to the decay rate of the freezing curve. We then applied a log-transformation on the observed freezing values and subsequently used ordinary least squares regression (log-linear model) for estimating the freezing curve parameters (intercept y(0) and decay rate b):

$$log_e(y(t)) = log_e(y(0)e^{-bt})$$

$$log_e(y(t)) = log_e(y(0)) + log_e(e^{-bt})$$

$$log_e(y(t)) = log_e(y(0)) - bt$$

The estimated intercept and decay rate are necessary and sufficient parameters to completely define the fitted freezing curve. Therefore, we used both measures as representative characteristics of our model for freezing behaviour to cluster animals into sustained and phasic responders. Additionally, we also included average freezing from bins 18–24 in the clustering model as this value is representative of the sustained component of the freezing curve. These parameters were chosen based on backward and forward greedy feature selection analyses with MR1 freezing parameters in 3 independent male and female cohorts as implemented in the clustvarsel R package to elucidate the most predictive variables. Average freezing bins 18–24 was identified as a predictive feature in all selection steps and decay rate and intercept of the fitted freezing curve were selected in 3 out of the 4 iterations. Subsequently, we fitted two-component Gaussian mixture models (GMMs) using the mclust package v5.4.10. Clustering

results were visualised with bivariate scatter plots showing individual points and 95% confidence ellipses around each cluster.

Pairwise comparisons between sustained and phasic responders were performed using unpaired t tests or Welch's t tests (in case of unequal variances between groups). Three groups (sustained, phasic, and shifters) were compared statistically using one-way analysis of variance (ANOVA) followed by Tukey's honestly significant difference test. Comparisons including multiple experimental factors but no repeated measures were facilitated with two-way ANOVA including main and interaction effects. Pairwise post hoc comparisons of model means were estimated using emmeans v1.7.4.1.

Differences in average freezing over time (repeated measures) between protocols, sexes, or phenotypes were investigated using linear mixed effects models as implemented in the lme4 package v1.1.29. Protocol/sex/phenotype, respectively, and time bin were included as fixed effects in the model and animal as a random effect. Pairwise comparisons of model means per time bin were calculated using emmeans v1.7.4.1.

Correlation matrices showing Pearson correlation coefficients were produced with the Hmisc package v4.7.0 and visualised using the corrplot package v0.92.

Unless otherwise stated, all p-values are two-tailed and a p-value $< = 0.05$ was considered statistically significant.

## RNA-seq data processing

Raw RNA-seq reads were trimmed for adapters and low-quality sequences using BBMap v38.86 [80]. Trimmed reads were then mapped against the UCSC mouse reference genome (build mm10) using the splice-aware aligner STAR v2.7.5a [81]. Gene-wise quantification of aligned reads was performed per sample using FeatureCounts from Subread v2.0.1 and the mm10 genome-associated gene annotation from UCSC [82,83]. The reference genome and annotation were both obtained from Illumina's iGenome database [84] (https://support.illumina.com/sequencing/sequencing_software/igenome.html). The quality of the raw data and aligned reads was comprehensively checked using fastQC v0.11.9 [85] and multiple functions of RSeQC v3.0.1 [86], respectively.

## Differential expression analysis

Significant gene expression differences between phasic and sustained responders were tested for each time point, gender, and brain region separately, by using the R package DESeq2 v1.34.0 [87–89]. All genes with a Benjamini–Hochberg adjusted p-value $<0.05$ were considered statistically significant.

## Functional enrichment analysis

For functional annotation of significant gene sets, we used the R package clusterProfiler v4.2.2 [90] to perform an overrepresentation test based on the hypergeometric distribution of terms from 3 databases, namely Gene Ontology [91,92], Kyoto Encyclopedia of Genes and Genomes [93], and REACTOME [94]. For cell type enrichment analysis, mouse-specific gene markers for major brain cell types, including astrocytes, microglial cells, oligodendrocytes, epithelial cells, and neurons, were extracted from the CellMarkers database [95] (downloaded on 03.05.2022) and used as input for the overrepresentation test function implemented in clusterProfiler. For male mice, we used all genes with an unadjusted p-value $<0.05$ as input for the overrepresentation analysis because of the low number of significant DEGs (adjusted p-value $<0.05$) and to see whether those genes give insights into biological differences between phasic and sustained responders that are comparable to those of female mice. The significance

threshold for overrepresented terms was set to a Benjamini–Hochberg adjusted p-value <0.05, and all expressed genes (genes remaining after filtering by DESeq2) of each group were used as background gene set, also known as universe.

### Disease overlap

To determine whether the differentially expressed genes between phasic and sustained freezing mice are associated with psychiatric disorders in humans, we extracted all human orthologs from the mouse DEGs using the function getLDS from the R package biomaRt v2.50.3 [96] and performed an intersection of those with differentially expressed genes between patients suffering from a psychiatric disorder including AAD, ASD, BD, SCZ, and PTSD and healthy controls published in 2 studies [51,52]. Since the human studies used a mixed sample of women and men for the differential expression analysis, we also combined DEGs from the comparison of sustained and phasic responders in both sexes for MR1. The intersection results were tested for significance using Fisher's exact test [97], and a significance threshold of $p < 0.05$ was used. Additional study information including the number of subjects, the percentage of women, the number of DEGs for each human disease as well as the overlapping genes in our study and in the human cohorts are included in S12 Table.

### Chord plot and circos plot visualisation of selected GO terms

The chord plot showing 15 overlapping DEGs between BLA MR1, BLA MR2, and CeA MR1 in female mice (Fig 3E) was generated using the GOPlot package v1.0.2. GO terms for biological processes of the gene set based on literature evidence were retrieved with a custom R script using the query function from the mygene package v1.32.0. Selected GO terms were grouped to higher-order categories manually based on semantic similarity (S10 Table).

The circos plot summarising significantly overrepresented GO terms for biological processes (Fig 4A) was generated with the circlize package v0.4.15. GO terms were manually assigned to higher level categories based on semantic similarity (S11 Table).

### Statistical analysis of electrophysiology experiments

Pairwise comparisons between sustained and phasic responders were performed using unpaired t tests or Welch's t tests (in case of unequal variances between groups). Individual variables were first screened for outliers by transforming raw values to z-scores. If a z-score exceeded the critical threshold of $|2.81|$ ($p < 0.005$, two-sided z-test), the corresponding measurement was removed from the statistical analysis due to being an outlier. Data were visualised using box plots.

## Supporting information

**S1 Fig. Influence of CS-US timing during training on CR during MR and sex-dependent differences in conditioned and unconditioned defensive behaviour. (a)** Schematic representation of the AAC protocol with predictable CS-US timing. **(b)** Schematic representation of the AAC protocol with unpredictable CS-US timing. **(c)** Average freezing curves of male mice during MR1 comparing groups conditioned with predictable (n = 12) and unpredictable (n = 17) protocols. **(d)** Phasic (green) and sustained (magenta) responders identified independently of CS-US timing during conditioning. **(e)** Female mice (n = 40) freeze significantly more than male mice (n = 40) during MR1 and MR2 throughout the later bins of CS presentation. **(f)** Female mice moved significantly more than males during the adaptation day. **(g, j)** Detailed analysis of extinction across multiple retrievals in both sexes. Phasic freezing (bins

13–17) underwent significant extinction from MR1 to MR6 in females (**g**) and males (**i**) as indicated by the negative coefficients in a linear regression model. At the same time, the extinction rate of the sustained component (bins 18–24) did not change significantly over time in either females (**h**) or males (**j**). (**k, l**) Histogram and density plots showing the bimodal distributions of freezing responses during bins 18–24 of MR1 for male (n = 40) and female (n = 40) mice. (**m**) No differences in freezing between males (n = 40) and females (n = 40) within the phasic (green) and sustained (magenta) phenotypes were detected during the first 30 s of CS presentation (bin 13). (**n**) Female sustained responders show a stronger bias to avoidance behaviour than males and female phasic responders during OFT, judged by the time spent in the periphery of the arena. All mice were conditioned using the protocol with unpredictable CS-US timing in e–j. *$p < 0.05$, **$p < 0.01$, ***$p < 0.001$, linear mixed effects model followed by pairwise comparison of model means between the predictable and unpredictable protocol (c), sustained and phasic groups within each protocol (d) and sexes within each MR in e, unpaired t test (f), two-way analysis of variance followed by pairwise comparison of model means between male and female mice within each respective phenotype (sustained and phasic) and phenotypes within sexes in m, and between sustained and phasic animals within each sex in n. Individual values underlying the experimental data are provided in the S1 Data excel file. AAC, auditory aversive conditioning; CS, conditioned stimulus; US, unconditioned stimulus; MR, memory retrieval; OFT, open field test.
(EPS)

**S2 Fig. Identification strategy of phasic and sustained responders for RNA-seq analysis after MR1. (a)** Schematic representation of the training protocol (AAC) timeline. **(b–d)** No systematic sex-dependent or phenotype-specific differences in freezing behaviour were found during training for the latency to the first freezing event **(b)**, freezing during the habituation phase **(c)**, and freezing during CS-ISI **(d)** (females n = 40, males n = 40). **(e)** Experimental timeline. **(f, g)** Clustering of male (n = 24) and female (n = 24) mice into phasic (green) and sustained (magenta) responders, and 95% confidence ellipses are drawn around each cluster. **(h, i)** Average freezing curves during MR1 for males (n = 24) and females (n = 24), black lines correspond to control animals not exposed to shock (n = 8). **(j)** Average freezing curves of animals selected for RNA-seq (n = 6 per group). All mice were conditioned using the protocol with unpredictable CS-US timing. *$p < 0.05$, **$p < 0.01$, ***$p < 0.001$, linear mixed effects model followed by pairwise comparison of model means between sustained and phasic freezers, female or male mice or training sessions in b–d and sustained and phasic freezers in h–j. Individual values underlying the experimental data are provided in the S1 Data excel file. AAC, auditory aversive conditioning; CS, conditioned stimulus; ISI, interstimulus interval; US, unconditioned stimulus; MR, memory retrieval.
(EPS)

**S3 Fig. Identification strategy of phasic and sustained responders for RNA-seq analysis after MR2. (a)** Experimental timeline. **(b)** Pairwise correlation between average freezing MR1 (bins 18–24) and MR2 (bins 18–24) in female mice (n = 40), r corresponds to the Pearson correlation coefficient. **(c, d)** Clustering of female mice into phasic (green) and sustained (magenta) responders based on MR1 and MR2 freezing response. Animals that were consistently assigned to a phasic or sustained phenotype in both retrievals, were further considered for transcriptomic profiling selection, the remaining animals were assigned to a shifter (grey) phenotype, as illustrated in **e** by the alluvial plot. **(f, g)** Individual (faded lines) and average (bold lines) freezing curves of phasic and sustained animals and shifters in MR1 and MR2. **(h)** Sustained animals spent significantly more time in the periphery of the arena in the OFT than phasic responders. **(i, j)** Individual (faded lines) and average (bold lines) freezing curves of

phasic (green) and sustained (magenta) responders selected for transcriptomic profiling (n = 6 per phenotype) during MR1 and MR2. **(k)** Average freezing (bins 18–24) during MR1 and MR2 of sustained and phasic animals selected for sequencing. **(l, m)** Female sustained responders selected for transcriptomic profiling spent significantly more time in the periphery of the arena in the OFT and had a tendency to spend less time in the light compartment during the LDT compared to the respective phasic animals. All mice were conditioned using the protocol with unpredictable CS-US timing. $*p < 0.05$, $**p < 0.01$, $***p < 0.001$, linear mixed effects model followed by pairwise comparison of model means between sustained and phasic freezers in f, g, i, j, and k, one-way analysis of variance followed by Tukey's post hoc test in h, unpaired t test in l and m. Individual values underlying the experimental data are provided in the S1 Data excel file. CS, conditioned stimulus; US, unconditioned stimulus; MR, memory retrieval; OFT, open field test.
(EPS)

**S4 Fig. Transcriptomic profiling of phasic and sustained responders. (a–f)** Volcano plots showing log2-fold change against adjusted p-value of all expressed genes in selected brain regions of phasic versus sustained responders. DEGs (adjusted p-value $<0.05$) are coloured in green and magenta for up- (LFC $> 0$) and down-regulation (LFC $< 0$) in phasic compared to sustained responders, respectively. **(g)** PCA plot capturing variance of gene expression in ACC and CeA of phasic and sustained males and females at early time-point (MR1). **(h)** PCA plot capturing variance of gene expression in ACC, BLA, CeA, and BNST of phasic and sustained females at early time point (MR1). **(i)** PCA plot capturing variance of gene expression in ACC, BLA, CeA, and BNST of phasic and sustained females at late time point (MR2). MR, memory retrieval; ACC, anterior cingulate cortex; BLA, basolateral amygdala; CeA, central amygdala; BNST, bed nucleus of stria terminalis; DEGs, differentially expressed genes; PCA, principal component analysis.
(EPS)

**S5 Fig. Isolation strategy of phasic and sustained responders for electrophysiological characterisation of LA principal neurons after MR2. (a)** Experimental timeline. **(b, c)** Clustering of female mice into phasic (green) and sustained (magenta) responders based on MR1 and MR2 freezing response (n = 38). Animals that were consistently assigned to a phasic or sustained phenotype in both retrievals, were further considered for the following electrophysiological characterisation, the remaining animals were assigned to a "shifter" (grey) phenotype. **(d, e)** Individual (faded lines) and average (bold lines) freezing curves of phasic (green), sustained (magenta) animals and "shifters" (grey) in MR1 and MR2. **(f, g)** Individual (faded lines) and average (bold lines) freezing curves of sustained (magenta) and phasic (green) responders selected for electrophysiological characterisation (n = 8 per phenotype) during MR1 and MR2. **(h)** Average freezing (bins 18–24) during MR1 and MR2 of sustained (magenta) and phasic (green) animals selected for electrophysiological characterisation. **(i)** Female sustained responders selected for electrophysiological characterisation showed a tendency to spend less time in the center of the arena in the OFT. **(j)** Phasic responders selected for electrophysiological characterisation spent significantly more time in the open arms and the extreme ends of the open arms **(k)** in the EPM test compared to the respective sustained animals. **(l)** Correlation matrix showing significant ($p < 0.05$) pairwise Pearson correlations between different behavioural outcome measures. All mice were conditioned using the protocol with unpredictable CS-US timing. $*p < 0.05$, $**p < 0.01$, $***p < 0.001$, linear mixed effects model followed by pairwise comparison of model means between sustained and phasic freezers in d–h, unpaired t test in i–k. Individual values underlying the experimental data are provided in the S1 Data excel file. CS, conditioned stimulus; US, unconditioned stimulus; MR, memory

retrieval; OFT, open field test; EPM, elevated plus maze test; LA, lateral amygdala.
(EPS)

**S6 Fig. Basic electrophysiological properties of lateral amygdala principal neurons.** Analysis of AP firing-related properties of primary neurons in acute coronal brain slices of the BLA of mice. No differences were found for **(a)** AP amplitude, **(b)** early, or **(c)** late adaptation of the APs, **(d)** after hyperpolarization and **(e)** sag of the recorded neurons of both phenotypes (n = 36/37 in phasic and n = 37/38 in sustained groups in a–e). Analysis of excitatory and inhibitory miniature events of primary neurons in acute coronal brain slices of the BLA of mice. No differences were detected in the decay tau for AMPAR-mediated mEPSCs **(f,** n = 24 in phasic and n = 20 in sustained groups**)**, nor GABAR-mediated mIPSCs **(g,** n = 23 in phasic and n = 17 in sustained groups**). (h)** Analysis of the excitation inhibition ratio (E/I ratio) of principal neurons of the BLA of both phenotypes. No difference was found for the E/I ratio based on the frequency of the excitatory and inhibitory miniature events recorded from primary neurons of the BLA of both phenotypes (n = 23 in phasic and n = 17 in sustained responders). Individual values underlying the experimental data are provided in the S1 Data excel file. BLA, basolateral amygdala; AP, action potential; mEPSC, miniature excitatory postsynaptic currents; mIPSC, miniature inhibitory postsynaptic currents.
(EPS)

**S1 Table. Differential expression results from different brain regions comparing phasic and sustained female mice 24 h (= MR1) and 28 days (= MR2) after fear conditioning.**
(XLSX)

**S2 Table. Differential expression results from different brain regions comparing phasic and sustained male mice 24 h after fear conditioning.**
(XLSX)

**S3 Table. Overrepresentation analysis results of functional terms using all significantly differentially expressed genes between female phasic and sustained responders 24 h and 28 days after fear conditioning (adjusted p-value < 0.05).**
(XLSX)

**S4 Table. Overrepresentation analysis results of functional terms using significantly up-regulated genes between female phasic and sustained responders 24 h and 28 days after fear conditioning (adjusted p-value < 0.05).**
(XLSX)

**S5 Table. Overrepresentation analysis results of functional terms using significantly down-regulated genes between female phasic and sustained responders 25 h and 28 days after fear conditioning.**
(XLSX)

**S6 Table. Overrepresentation analysis results of functional terms using all genes with an unadjusted p-value < 0.05 between male phasic and sustained responders 24 h after fear conditioning.**
(XLSX)

**S7 Table. Overrepresentation analysis results of functional terms using up-regulated genes with an unadjusted p-value < 0.05 between male phasic and sustained responders 24 h after fear conditioning.**
(XLSX)

**S8 Table. Overrepresentation analysis results of functional terms using down-regulated genes with an unadjusted p-value < 0.05 between male phasic and sustained responders 24 h after fear conditioning.**
(XLSX)

**S9 Table. List of experimental animals and batches used in the study.**
(XLSX)

**S10 Table. GO terms for Chord Plot related to Fig 3E.**
(XLSX)

**S11 Table. GO terms for Circos Plot related to Fig 4A.**
(XLSX)

**S12 Table. Human psychiatric disorder-related genes and overlap with the differentially expressed genes in this study related to Fig 3I.**
(XLSX)

**S1 Box. Contains definitions of key terms used in the paper.**
(DOCX)

**S1 Data. Individual values underlying the data presented in Figs 1, 2, 5, S1, S2, S3, S5, and S6.**
(XLSX)

## Acknowledgments

Authors thank StarSEQ GmbH, Mainz for help in the selection of optimal sequencing parameters. Authors thank Andrea Conrad and Mona Flachsel for their help with organisation and performance of animal experiments. We would also like to thank Michael Plenikowski for valuable advice in graphic design of figures.

## Author Contributions

**Conceptualization:** Irina Kovlyagina, Hristo Todorov, Margarita Tevosian, Beat Lutz.

**Data curation:** Irina Kovlyagina, Anna Wierczeiko, Hristo Todorov.

**Formal analysis:** Irina Kovlyagina, Anna Wierczeiko, Hristo Todorov, Eric Jacobi.

**Funding acquisition:** Jakob von Engelhardt, Susanne Gerber, Beat Lutz.

**Investigation:** Irina Kovlyagina, Anna Wierczeiko, Hristo Todorov, Eric Jacobi, Margarita Tevosian, Jakob von Engelhardt, Susanne Gerber, Beat Lutz.

**Methodology:** Irina Kovlyagina, Eric Jacobi, Margarita Tevosian.

**Project administration:** Irina Kovlyagina, Beat Lutz.

**Resources:** Jakob von Engelhardt, Susanne Gerber, Beat Lutz.

**Supervision:** Jakob von Engelhardt, Susanne Gerber, Beat Lutz.

**Visualization:** Irina Kovlyagina, Anna Wierczeiko, Hristo Todorov.

**Writing – original draft:** Irina Kovlyagina, Hristo Todorov.

**Writing – review & editing:** Irina Kovlyagina, Anna Wierczeiko, Hristo Todorov, Eric Jacobi, Margarita Tevosian, Jakob von Engelhardt, Susanne Gerber, Beat Lutz.

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
