## [Editor Report · Decision Letter 0]

21 Mar 2024

Dear Beat, 

Thank you for submitting your manuscript entitled "Freezing responses during prolonged threat memory retrieval reflect trait-like anxiety  endophenotypes in female and male inbred mice" for consideration as a Methods and Resources by PLOS Biology.

Your manuscript has now been evaluated by the PLOS Biology editorial staff and the academic editor and I am writing to let you know that we would like to move forward with your manuscript.

However, before we can do so, we need you to complete your submission by providing the metadata that is required for full assessment. To this end, please login to Editorial Manager where you will find the paper in the 'Submissions Needing Revisions' folder on your homepage. Please click 'Revise Submission' from the Action Links and complete all additional questions in the submission questionnaire.

Once your full submission is complete, your paper will undergo a series of checks. To provide the metadata for your submission, please Login to Editorial Manager (https://www.editorialmanager.com/pbiology) within two working days, i.e. by Mar 23 2024 11:59PM.

Kind regards,

Christian

Christian Schnell, PhD

Senior Editor

PLOS Biology

cschnell@plos.org

---

## [Editor Report · Decision Letter 1]

28 Mar 2024

Dear Beat,

Thank you for your patience while we considered your revised manuscript "Freezing responses during prolonged threat memory retrieval reflect trait-like anxiety  endophenotypes in female and male inbred mice" for publication as a Methods and Resources at PLOS Biology. This revised version of your manuscript has been evaluated by the PLOS Biology editors and the Academic Editor.

Based on our Academic Editor's assessment of your revision, we are likely to accept this manuscript for publication, provided you satisfactorily address the following data and other policy-related requests:

* We would like to suggest a different title to improve readability: "Assessment of freezing responses during prolonged threat memory retrieval enables the study of trait-like anxiety and stress susceptibility in female and male mice"

* Please add the links to the funding agencies in the Financial Disclosure statement in the manuscript details.

* Please include the full name of the IACUC/ethics committee that reviewed and approved the animal care and use protocol/permit/project license. Please also include an approval number.

* Please note that per journal policy, we do not allow the mention of "data not shown", "personal communication", "manuscript in preparation" or other references to data that is not publicly available or contained within this manuscript. Please either remove mention of these data (line 527) or provide figures presenting the results and the data underlying the figure(s).

DATA POLICY:

Regardless of the method selected, please ensure that you provide the individual numerical values that underlie the summary data displayed in the following figure panels as they are essential for readers to assess your analysis and to reproduce it: 1G, 2G, 2H, 2J, 2K, 2M, 2N, 5C–G, 5I–O, and similar panels in the supplementary figures.

CODE POLICY

Per journal policy, if you have generated any custom code during the curse of this investigation, please make it available without restrictions upon publication. Please ensure that the code is sufficiently well documented and reusable, and that your Data Statement in the Editorial Manager submission system accurately describes where your code can be found. 

We expect to receive your revised manuscript within two weeks. 

*Published Peer Review History*

*Press*

Sincerely,

Christian

Christian Schnell, PhD

Senior Editor

cschnell@plos.org

---

## [Editor Report · Decision Letter 2]

25 Apr 2024

Dear Beat,

Thank you for the submission of your revised Methods and Resources "Leveraging inter-individual variability in threat conditioning of inbred mice to model trait anxiety" for publication in PLOS Biology. On behalf of my colleagues and the Academic Editor, Eric Nestler, I am pleased to say that we can in principle accept your manuscript for publication, provided you address any remaining formatting and reporting issues. These will be detailed in an email you should receive within 2-3 business days from our colleagues in the journal operations team; no action is required from you until then. Please note that we will not be able to formally accept your manuscript and schedule it for publication until you have completed any requested changes.

PRESS

Sincerely, 

Christian

Christian Schnell, PhD

Senior Editor

PLOS Biology

cschnell@plos.org